# Coaxial Cable Distributed Strain Sensing: Methods, Applications and Challenges

**DOI:** 10.3390/s25030650

**Published:** 2025-01-22

**Authors:** Stephanie King, Gbanaibolou Jombo, Oluyomi Simpson, Wenbo Duan, Adrian Bowles

**Affiliations:** 13-Sci Limited, Hampshire PO13 9FU, UK; arbowles@3-sci.com; 2Centre for Engineering Research, School of Physics, Engineering and Computer Science, University of Hertfordshire, Hatfield AL10 9BA, UK; o.simpson@herts.ac.uk (O.S.); w.duan@herts.ac.uk (W.D.)

**Keywords:** distributed sensing, structural health monitoring, coaxial cable Fabry–Perot interferometry, time domain reflectometry, coaxial cable Bragg grating

## Abstract

Distributed strain sensing is a powerful tool for in situ structural health monitoring for a wide range of critical engineering infrastructures. Strain information from a single sensing device can be captured from multiple locations simultaneously, offering a reduction in hardware, wiring, installation costs, and signal analysis complexity. Fiber optic distributed strain sensors have been the widely adopted approach in this field, but their use is limited to lower strain applications due to the fragile nature of silica fiber. Coaxial cable sensors offer a robust structure that can be adapted into a distributed strain sensor. They can withstand greater strain events and offer greater resilience in harsh environments. This paper presents the developments in methodology for coaxial cable distributed strain sensors. It explores the two main approaches of coaxial cable distributed strain sensing such as time domain reflectometry and frequency domain reflectometry with applications. Furthermore, this paper highlights further areas of research challenges in this field, such as the deconvolution of strain and temperature effects from coaxial cable distributed strain sensor measurements, mitigating the effect of dielectric permittivity on the accuracy of strain measurements, addressing manufacturing challenges with the partial reflectors for a robust coaxial cable sensor, and the adoption of data-driven analysis techniques for interrogating the interferogram to eliminate concomitant measurement effects with respect to temperature, dielectric permittivity, and signal-to-noise ratio, amongst others

## 1. Introduction

Data on parameters such as strain provide reassurance that structural assets are working within their design limits and provide an early warning if safe levels are exceeded [1]. This information enables structural failure avoidance, which can prevent heavy human and financial costs [2]. Structural health monitoring (SHM) plays a critical role across a breadth of industries, such as civil infrastructure, energy production, aerospace, and automotive, in the management of existing and future structures [3]. Growth industries such as offshore wind present new challenges for SHM due to the range of operating environments and structural requirements on the various turbine components [4]. Information gathered by the Caithness Windfarm Information Forum 2014 [5] reported that there is a clear trend in the number of accidents increasing as more wind turbines are being built. Blade failure and structural failure were cited as two of the main causes of accidents.

As well as avoiding accidents by pre-empting failures, in situ condition monitoring decreases the volume of routine manual inspection work that is often carried out in exposed environments with awkward access. Reducing the regularity of human interaction with the structure would be a secondary route to lowering accident rates. Remotely tracking the operation and health of assets can feed into scheduling risk-based inspections where inspections and maintenance efforts are focused on areas deemed to carry the highest risk. This optimizes the time and cost efficiency of operation and maintenance (O&M) labor, which is vital for remote, difficult-to-access assets exposed to challenging environmental conditions [6].

Furthermore, real-time, in situ condition monitoring can be used to optimize the operation of the assets, ensuring that the system is not driven beyond limits which could incite structural ‘wear and tear’. This extends the lifetime of the structures and ultimately reduces the need to prematurely dispose of components, minimizing costs, and material waste detrimental to the planet.

Structural health monitoring is an extensive field of research and technology development. In the context of trains sensing, there are several incumbent and emerging technologies summarized in Table 1. This summarized information provides a broader perspective to set the scene for how optical fiber and coaxial cable strain sensing technologies, the key focus of this paper, compare with other strain sensing techniques.

It becomes impractical to use arrays of discrete sensors to measure strains over large or extensive structures due to the costs of installation, wiring, and analysis of a multitude of sensor devices [7]. Also, discrete sensors measure localized events and can miss important behaviors occurring on non-instrumented sections of the structure [10].

Fiber optic technology can provide distributed strain data, where a single device (the fiber optic cable) can report on the conditions at continuous locations over an entire length. Due to the relatively low strain limit of the silica in fiber optics, there is a limit on the magnitude of strain this technology can monitor [15,21]. Fiber optic technology is therefore not capable of capturing high-strain events. There is also a challenge around the installation of fiber optics. The fragile fibers must be handled carefully, which presents challenges during installation procedures and operation in exposed or extreme conditions [10]. However, there are techniques used in fiber optic sensing that can be applied to other media such as coaxial cables [15,21,22,23].

Coaxial cables are more robust than fiber optics and can also provide distributed strain information. There has been significant interest and development in the field of coaxial cable distributed sensing in recent years [16]. This paper outlines the development of coaxial cable distributed sensing for strain applications, from the origins of fiber optic sensing techniques through to time domain and frequency domain methods employed on coaxial cables, and reviews the latest developments in coaxial cables as distributed strain sensors.

The development of coaxial cable distributed sensing is, to some extent, analogous to the development of fiber optic distributed sensing. Figure 1 illustrates the similarities in the two fields and shows the structure of this paper.

This paper provides an overview of the research progress in the development and application of distributed strain sensing with a specific focus on coaxial cable sensors. To enable the reader to navigate this paper efficiently, it has been split into the following sections:Section 2. Background on fiber optical distributed sensing:
○Section 2.1. Time domain reflectometry in fiber optical sensing;○Section 2.2. Frequency domain reflectometry in fiber optical sensing.Section 3. Coaxial cable distributed strain sensing:
○Section 3.1. Time domain reflectometry in coaxial cable distributed strain sensing;○Section 3.2. Frequency domain reflectometry in coaxial cable distributed sensing.Section 4. Future research challenges in frequency domain coaxial cable strain sensing.

## 2. Background on Fiber Optical Distributed Sensing

The development of extremely low-loss optical fibers in the late 1970s opened up the field of optical fiber sensors (OFSs). By 1982, a range of sensing applications were under research and development, including magnetic, acoustic, temperature, and strain, among others [10]. The applications are wide-ranging, including seismology [24], biomechanics [25], shape sensing [26], and structural health monitoring (SHM) [2]. Details on the working principles and applications of fiber optic sensors are well documented [27]. There are three main branches of OFSs: grating-based sensors, interferometric sensors, and distributed sensors, as described in Figure 2.

As a very basic description, time and frequency domain reflectometry-based fiber optical sensing works by sending a pulse of light along a fiber and collecting the reflected light signal. An image of a distributed fiber optic sensor is shown in Figure 3. Optical fibers are bonded to, or embedded in, the structure of interest. A change in strain in the structure transfers to the optical fiber, altering the way in which the light is reflected. By studying the reflected light, it is therefore possible to infer the change in strain [10]. Some of these techniques utilize time domain reflectometry (TDR), whereby the arrival times of the reflected light are analyzed. The alternative is frequency domain reflectometry (FDR), whereby the frequencies of the reflected light are used to interpret strain events in the structure. Figure 1 describes the classification of the different fiber optical sensors into time domain or frequency domain techniques [8].

### 2.1. Time Domain Reflectometry in Fiber Optical Sensing

Scattering-based techniques (Rayleigh scattering, Raman scattering, and Brillouin scattering) are used for time domain reflectometry in optical fibers [28]. The entire continuous length of the fiber optic cable is turned into a sensor, achieving truly distributed sensing down to approximately 1 mm spatial resolution [8]. Scattering is the process by which the directional energy in a propagating wave is transferred to other directions. Light undergoing linear scattering has no change in frequency, whereas light experiencing nonlinear scattering will undergo a frequency shift.

Optical time domain reflectometry (OTDR) is a technique that was developed in the 1980s to test optical telecommunication fibers. Essentially, a short optical pulse is transmitted down the fiber, and the backscatter is monitored, which provides information on the condition of the fiber from which environmental conditions external to the fiber, but acting on it, can be inferred. This method was developed for all three types of scattering (Raman, Rayleigh, Brillouin), with each providing different advantages and disadvantages.

Rayleigh scattering describes the elastic scattering of light by inhomogeneities much smaller than the wavelength of the incident light [29,30]. Under normal conditions in a fiber optic, as light interacts with the fiber, scattered light remains at an angle that supports forwards propagation. On interaction with a discontinuity, some of the light will be scattered at an angle which does not support forwards propagation, or scattered backwards, towards the light source, which is the principle behind OTDR. Rayleigh scattering is used to analyze attenuation associated with breaks, splices, connectors, and the general health of a fiber [2,31]. Other applications explored include measuring landslide activity [32] and detecting cracks in concrete structures [33]. This technique offers the highest spatial resolution of the three scattering methods [8], but it is highly sensitive to vibrations [34], so it also finds applications in acoustic sensing. The sensing range of Rayleigh scattering is limited to around 70 m.Raman scattering is nonlinear and arises from the interaction between light and the vibrational and rotational transitions of the atoms within the fiber material structure. Depending on the exact transitions, the scattered light will shift in frequency to two discrete bands; anti-Stokes transmission describes the shift to a higher frequency, and Stokes transmission describes the shift to a lower frequency. The ratio of the magnitude of these peaks provides data on the temperature of the fiber [2]. This technique provides temperature information independent of strain, but it does have a poor signal-to-noise-ratio [35].Brillouin scattering is nonlinear and associated with electrostriction, which couples electromagnetic waves with material structure-scale waves and phonons. Incident electromagnetic energy interacts with the optical fiber material to create backscattered electromagnetic energy of a lower frequency and a phonon of a low-frequency vibrational energy. The thermal energy within the fiber will influence the Brillouin scattering [29,30]. This technique is used for temperature sensing and also strain sensing (if a separate temperature measurement independent of strain can be taken, enabling strain to be inferred). One notable advantage, in the context of this paper, is the development of Brillouin optical time domain reflectometry (BOTDR), through which monitoring could be achieved from one end of the fiber. Single-ended sensing is an attractive practical advantage when considering the deployment of distributed sensors in engineering structures [10]. This has led to Brillouin scattering being the most common technique used in civil applications of structural health monitoring [36]. This method enables a long measurement range (kilometers (kms)) but with limitations on the spatial resolution to around 1 m [8].

The trade-off between spatial resolution and range drove the development of the techniques and then led to Optical Frequency Domain Reflectometry (OFDR). The equipment required to obtain high spatial resolutions with OTDR techniques is complex and therefore expensive; a narrow light pulse is necessary, but this generates a poor signal-to-noise ratio, which necessitates a receiver with an increased bandwidth for any signal detection [10]. Huang et al. [23] also cites the poor signal-to-noise ratio (SNR) as a limitation of OTDR. In order to address these disadvantages, frequency domain reflectometry was developed for fiber optical sensing.

### 2.2. Frequency Domain Reflectometry in Fiber Optical Sensing

In OFDR, the backscatter is monitored in the frequency domain; the interference fringes of the backscattered signals are monitored, and a shift in the interference pattern infers an environmental change that can be located through the phase information of the frequency signal. This technique is employed for grating-based sensors, interferometric sensors, and some scattering-based sensors.

#### 2.2.1. Scattering-Based Sensors

The natural impurities in the silica of the fiber optic give rise to an inherent variation in the refractive index along the length of the fiber. Rayleigh scattering occurs at each variation in the refractive index and gives each optical fiber a unique ‘fingerprint’ from which changes can be monitored. Raman scattering is a result of incident light interacting with the rotational and vibrational atomic transitions within the fiber optic material. Brillouin scattering is the interaction of incident light with the larger-scale material structure vibrational modes of the fiber.

The three scattering mechanisms (Raman, Rayleigh, Brillouin) all offer different advantages and disadvantages, as outlined in Section 2.1. Recent work has explored the possibility of combining all three techniques for an optimized solution [37].

#### 2.2.2. Grating-Based Sensors

Grating-based sensors such as Fiber Bragg Gratings (FBGs) enable parameters such as strain to be measured along fiber optic cables. Periodic modifications in the refractive index of the fiber in the axial direction create narrow-band reflections at discrete resonant frequencies [21]. The exact resonant frequencies can be monitored. A change in environmental conditions will change the optical length of the grating features, shifting the resonances.

FBGs can either be used as point sensors or quasi-distributed sensors through the ability to multiplex gratings along the fiber length; several gratings of different periods can be inscribed on a single fiber [2,11]. The strain sensing capability of FBGs has made them suitable for a wide range of applications: strain analysis of power transmission lines [38], soil strain sensing [39], corrosion sensor based on strain measurements [40], and leakage monitoring via hoop strain sensing [41].

#### 2.2.3. Interferometric Sensors

Optical interferometric techniques (Fabry–Perot, Mach–Zehnder, Sagnac) all work on the principle of recombining two optical signals that have experienced different optical paths and analyzing the resultant interference pattern [42]. Each technique uses a different arrangement to generate the two optical beams [43,44]. Fabry–Perot interferometry holds the advantage that it is an in-line arrangement, so it can be employed on a single cable. Other interferometric techniques utilize two cables: one experiencing environmental change, the other remaining in steady-state acting as a reference. The Surveillance d’Ouvrages par Fibres Optiques technique is one of the more successful approaches to obtaining integrated measurements; a single elongation value is determined from the integration of behavior along the length of the fiber [2].

Whilst the OFDR methods are more cost-effective and achieve higher spatial resolution than the OTDR techniques, the sensing range is more limited. Table 2 summarizes the advantages and disadvantages of the various distributed optical fiber sensors techniques.

Fiber optics distributed sensing finds applications in monitoring buildings, bridges, tunnels and roads, crack growth in concrete structures, strain measurements in wind turbine blades and pile foundations, monitoring land slope stability, soil and rock deformations and monitoring the stability of ground anchors, oil and gas pipeline degradation, and the detection of failure in hydraulic engineering structures, amongst others, as illustrated in Figure 4 [8,10].

However, fiber optics distributed strain sensing presents the following challenges:The inherent weakness of the fiber optic material limits its application to lower strain events. In order to realize a wider range of measurable strain and monitor structures up to failure, a more robust ‘carrier’ would be necessary [10,45].There are several references quoting the strain limits of optical fibers:
○10,000 µε (1%) [7,9];○4000 µε (0.4%) [21,46].The fragility of the fiber optics makes installation difficult, and great care must be taken not to damage the sensor itself. Bending stresses should be avoided during installation as this can impact the weakly scattered signals necessary for monitoring [10].There is concern that the fragility implies a strong limitation on the long-term exposure to harsh environments [10,21,45].The expense of fiber optic systems can limit their use to applications only where reliability is critical [2,47]. A more cost-effective solution could see SHM techniques applied where reliability or safety is less critical but nonetheless would benefit from the economic advantages in targeted O&M activities, or the optimization of system operating parameters.

## 3. Coaxial Cable Distributed Strain Sensing

Coaxial cables and optical fibers follow similar principles of electromagnetic theory and signal propagation, albeit at different frequencies of the electromagnetic spectrum. Similar techniques developed for DOFSs can be applied to the medium of coaxial cables. However, coaxial cables are advantageous as they are more robust, can withstand larger strain, and present a lower cost [15,21,45,46]. The coaxial configuration of the conductors also provides shielding from electromagnetic interference [48,49].

The background of the development of coaxial cables as distributed strain sensors follows an analogous path to that already outlined for fiber optics, and this is presented in the following Section 3.1 and Section 3.2.

### 3.1. Coaxial Cable Time Domain Reflectometry

Time domain reflectometry for fault location is a well-established technique [50]. A short pulse signal propagating along a waveguide will create a reflection at an impedance discontinuity (change in cross-sectional area) caused by a physical fault [51,52]. The arrival time of this reflection identifies the location of the fault. The principle of this technique is depicted in Figure 5.

For a lossless transmission line, the characteristic impedance Z_0_ is described as follows:(1)$$Z_{0}=\sqrt{\frac{L}{C}}=\frac{1}{2\pi }\sqrt{\frac{\mu_{0}\mu_{r}}{\varepsilon_{0}\varepsilon_{r}}}\ln\left(\frac{D_{o}}{D_{i}}\right),$$
where Z_0_ is the characteristic impedance of the cable, L is the inductance per unit length, and C is the capacitance per unit length. The properties of the dielectric are given by ε_0_ the permittivity of free space, ε_r_ the relative permittivity of the dielectric in the coaxial cable, µ_0_ the permeability of free space, and µ_r_ the permeability of the dielectric in the coaxial cable. The cable dimensions are D_i_ the outer diameter of the inner conductor and D_o_ the inner diameter of the outer conductor.

Varying any of these parameters will change the impedance Z_0_. Where there is an impedance discontinuity, a reflection will occur in accordance with the below equation:(2)$$\rho =\frac{Z_{0}-Z_{1}}{Z_{0}+Z_{1}}\;,$$
where ρ is the reflection coefficient, Z_0_ is the impedance before the discontinuity, and Z_1_ is the impedance after the discontinuity. If Z_0_ is different to Z_1_ (in a region where the material properties or physical geometries differ), then a reflection will occur.

This technique has been used to monitor the health of cables themselves and identify wiring faults [13,52,54]. The ability of TDR to locate geological discontinuities has been reported, where changes and fractures in rock following a sub-terranean nuclear explosion were monitored by way of monitoring the change in unbroken cable length [55]. Initially, coaxial cable TDR could only monitor events and not provide measurements due to challenges relating the TDR reflections to the true deformation. Work has been undertaken to quantify the relationships between shear deformation and change in impedance, axial and transverse loads and impedance change, and axial strain and reflected voltage, as well as assess the accuracy of using TDR to measure impedance [48,56,57,58,59]. This work means that the TDR method could be extended to infer environmental conditions external to, but impacting on, the cable, giving rise to localized impedance variations [56]. Monitoring the reflections provides information on the location of damage or environmental change acting on the cable, such as strain [60]. In this way, coaxial cables can become distributed sensors using TDR to interrogate the cable.

A number of papers cite the use of coaxial cables as distributed structural health sensors using TDR in applications such as monitoring the integrity of large-diameter wire rope [61], rock and soil movement in mining activities [22,61], the prediction of slope failure in open-cast mines [51], landslide monitoring [62], bridge scour monitoring [63,64], and crack detection in reinforced concrete structures such as bridges and buildings [7,47,65].

A challenge associated with using TDR for distributed strain sensing includes the lack of sensitivity of the technique [47,56,65]. There is also an associated difficulty in ascertaining the relationship between the voltage of reflected signals and the associated impedance change when the impedance has changed gradually over a length of cable and is not a sharp discontinuity [56]. The short pulse required for TDR spectrally spreads with transmitted distance, reducing the accuracy of the technique over long distances [52,54,66,67,68].

Coaxial cable time domain reflectometry is a single-ended technique that can be employed on any coaxial cable without the need for modifications to that cable. It can therefore be used to analyze an existing, in situ cable from any point in the cable’s lifetime without prior requirement to anticipate future TDR measurements. The potential of the technique has been pursued since 1971 [55]. A summary of published work reporting the most recent advances in TDR for strain sensing/mechanical movement monitoring using unmodified coaxial cables is provided in Section 3.1.1.

#### 3.1.1. TDR Using Unmodified Coaxial Cables

Guidelines to optimize TDR for landslide monitoring were deduced in an effort to encourage the adoption of this technique to geological applications [62]. The scenario under investigation is illustrated in Figure 6. Three types of coaxial cable were tested: RG-8, P3-500CA, and P3-500 JCASS in a custom-built shear box designed to simulate landslide events. Various soil types and cable grout arrangements were tested. Removal of the cable jacket is advised to avoid slippage, which reduces the sensitivity of the technique, but the increased exposure to corrosion was noted. A standardized grout mixture was recommended to simplify installation and also aid signal interpretation as the known grout interaction could be accounted for via signal analysis.

This study built on work published in where TDR was investigated to predict slope failure in open-cast mines [51]. This research involved testing two different coaxial cable types, RG-6 and RG-213, first in lab-based equipment and then in a 5-month field trial at the Manganese Ore India Limited Dongri Buzurg mines. The experimental tests related the reflection coefficient of the TDR signal to the shear deformation of the cable. For RG-6, the average highest deformity by shear failure was 11 mm, and for RG213, it was 14 mm. There was insufficient ground movement in the field trials to register a notable change in the reflection coefficient of the TDR response. RG-6 was recommended due to its increased sensitivity and cost-effectiveness.

Recent work has shown that random reflections intrinsic to an unmodified coaxial cable can be tracked to monitor more subtle changes in environmental conditions [22,70]. Whilst this work focused on tracking changes in temperature using random reflections on an unmodified coaxial cable, the principle is analogous when applied to tracking changes in strain. Essentially, a change in temperature causes a change in physical distance between sites of random reflections due to the coefficient of the thermal expansion of the materials in the cable. The electrical path length would also change with temperature due to the change in dielectric permittivity. Similarly, a change in strain would cause a change in physical distance between sites of random reflections, and it would also cause a change in the electrical path length due to the photoelastic effect altering the permittivity of the dielectric with strain [46]. A novel method using the random reflections of a coaxial cable to enable temperature monitoring in this way is presented in [22]. A cross-correlation analysis of TDR S11 signals before and after heating was applied to establish the location and magnitude of a temperature change along a 14 m length of the RG58 cable. The cross-correlation identifies a shift in the S11 signal using the random reflections as position markers to align the S11 signals before and after heating. A misalignment in the signals indicates a change in temperature through a change in the arrival time of the random reflections. An analogous result could be achieved through the application of strain instead of temperature.

Fundamental research into using a coaxial cable as a distributed strain sensor using TDR was conducted using the standard RG-174 coaxial cable. A standard RG-174 coaxial cable was subjected to a localized lateral compression at a fixed distance. The TDR signal of the uncompressed cable was first baselined so that the impact of the application of the lateral force could clearly be observed. This loading condition resulted in a sharp impedance change at the location of the applied force. The cable was also subjected to axial tension tests with a fixed section of the cable being stretched in controlled increments. The increased tensile strain reduced the cross-sectional area of the cable, decreasing the impedance of the stretched cable section and increasing the reflected voltage level. A prototype sensor, similar in geometry and size to RG-174 but with a rubber dielectric, was tested alongside the RG-174 cable for comparison. The prototype sensor (diagram shown in Figure 7) demonstrated the greater sensitivity of the applied loads compared with the RG-174 cable due to the more compliant dielectric material [48]. The level of random noise in the TDR measurements was noted and, whilst the load levels tested were clearly visible above the noise level, could be problematic when monitoring lower levels of strain.

These practical works build on an analytical study that was conducted to assess the theoretical possibility of using TDR to purpose a coaxial cable as a distributed strain sensor [56]. The purpose of the analysis was to ascertain a relationship between reflected voltage measured through TDR and axial strain. Without this type of investigation, and other similar work conducted on deducing direct relationships between TDR response and shear and tensile deformations, coaxial cable TDR could only monitor events, not provide direct measurements of strain or loading conditions [48,57,58].

#### 3.1.2. TDR Using Modified Coaxial Cables

There are several ways in which coaxial cables have been modified in the pursuit of coaxial cable strain/mechanical movement sensing using TDR. The intended purpose of the different modifications varies and is summarized in Table 3.
Lack of sensitivity using coaxial cable for strain sensing remains a challenge using TDR [47,56,65]. As described earlier in this paper, reviewing the progress of the unmodified coaxial cable, Lin et al. [48] explored the use of a more compliant rubber as the dielectric in a coaxial cable structure by way of increasing the sensitivity to strain. The sensitivity of a prototype rubber-based dielectric cable was shown, through experiments, to be approximately five to ten times that of the standard RG-174 coaxial cable. This approach still relies on a geometric change with strain, whereas a change in the topology of the outer coaxial cable conductor with strain could inherently offer increased sensitivity.Several published reports describe the use of helical wound outer conductors as a technique to induce a change in outer conductor topology with strain [7,9,47,71,72]. A lot of this work focused on crack detection in reinforced concrete beams. In 2004, Chen et al. tested a prototype coaxial cable sensor constructed with helically wound adhesive copper tape forming the outer conductor (Figure 8) [7].

When compared with the coaxial cable with a regular copper braided outer conductor, it was found that the prototype sensors were 15–80 times more sensitive than sensors based on an off-the-shelf coaxial cable and could offer a spatial resolution of 50 mm. This work verified that the change in the topology of the outer conductor has a greater impact on sensitivity to strain than relying on a geometric change. Subsequent developments to improve consistency in the sensor performance of this approach included the inclusion of a Teflon dielectric with a commercial steel spiral wrapper covered with a thin solder layer to improve electrical continuity between adjacent spirals [71]; the replacement of the solder layer with a plasma-sprayed coating to create a more uniform, reproducible sensor [47]; performance validation of the copper tape spiral and Teflon/steel spiral devices through field trials in a highway bridge over a 5-year period [72] (Figure 9) and replacement of the spiral-wound outer conductor with a solid outer conductor inscribed with a shallow helical groove [9]; and with the purpose of maintaining the topographic contribution to strain sensitivity but avoiding the problem of poor electrical continuity seen in spiral-wrap configurations. This impacted signal attenuation and limited the length of the sensors. The results from this work suggested that this sensor design could detect cracks of 0.02 mm and identify multiple cracks with a minimum of 3 mm spatial resolution, although further work was noted regarding the assessment of the improved signal attenuation of spiral-wound topologies [9].

Through winding two parallel conductors helically around a central silicone core, a different approach to harnessing the change in topology to increase the TDR coaxial cable’s sensitivity to strain was researched [60,74]. This alternative configuration is illustrated in Figure 10. The work concluded that this different ‘coaxial’ design could be used for the distributed monitoring of deformation.
The techniques described above rely on the creation of a new reflection point being generated, alerting the user to a source of increased strain or mechanical movement. A different approach, whereby a coaxial cable is crimped prior to installation, is referred to in [61]. The crimps create reflections at known distances along the cable, so they act as location reference points to improve the spatial accuracy of CCTDR. Crimping coaxial cables to improve the location tracking of ground movements are also mentioned in the 2020 research into investigating TDR as a means of predicting slope failure in open-cast mines [51]. ‘Denting’ the coaxial cable to increase the SNR of reflected signals, making it easier to track with changing conditions, was mentioned in [70] but dismissed due to the impact on the mechanical strength of the cable.More extreme modifications of cables to enhance the distributed strain sensing using TDR have involved the inclusion of discreet sensors along the cable. Novel sensing elements created from piezoresistive multi-walled carbon nanotubes were incorporated along the standard speaker cable. Separate lengths of the speaker wire were joined via one or more sensing elements, with a reflection in the time domain occurring at these inclusions (Figure 11). The nanocomposite exhibits an increased response to strain, which could enhance the sensitivity of the technique. Challenges were faced with multiple sensing elements where signal attenuation and reflections at prior sensing elements resulted in very small signal levels in the sensing elements at the end of the cable [53].

A similar idea was evaluated in the context of incorporating capacitive tilt sensors along a coaxial cable for TDR measurements to monitor ‘crosslevel’ or difference in height between adjacent train tracks. The device tested is shown in Figure 12, with a single in-line sensor present. This was scaled up to test up to three in-line sensors. Ground movements can cause the top surface of the tracks to come out of alignment, causing derailments. Testing validated the operation of the tilt sensors themselves and confirmed that TDR accurately measured the physical location of the tilt sensors. TDR could be used to analyze in-line tilt sensors, but again challenges were faced with multiple sensors and signal degradation with additional sensors [73].

There are some remaining challenges with the TDR method for distributed sensing that could not be fully addressed by modifying the design of the coaxial cable.

Poor SNR and low sensitivity limit the technique [9,56,75].Multiple discontinuities give rise to secondary reflections, complicating signal deconstruction for analysis [58].Cable attenuation and energy lost through partial reflection at each impedance change result in the degradation of signal clarity for additional faults along the cable [76].Spatial resolution degenerates along the cable as the excitation signal distorts due to frequency-dependent cable attenuation [52,54,58,66,76].

To overcome the shortcomings of TDR, a multitude of signal analysis techniques have been developed. These aim to refine the TDR technique beyond the advances previously described, which were achieved through a modification of the physical cable design.

A reconstruction method, developed to overcome the limitation of inaccuracies when multiple discontinuities are present on a TDR system, was proposed [77]. The impedance of a transmission line was reconstructed from the waveshape of a reflected signal by dividing the reflected wave into equal subintervals for analysis. This model did not account for dispersion or loss.

State-of-the-art signal analysis techniques include spread spectrum time domain reflectometry, noise domain reflectometry, and their derivatives. An overview of these is explained well in [52,54,78]. State-of-the-art systems such as Viper Innovations products [79] utilize some of these methods. The key objective of these techniques is to enable monitoring on live cables and in high-noise environments. Joint time–frequency analysis (JTFA) or time–frequency domain reflectometry (TFDR) bring an added advantage to analyzing multiple faults along a cable [76]. Conventional TDR struggles to detect multiple faults due to the inherent frequency-dependent attenuation of coaxial cables, distorting the signals of more distant faults. The partial reflection of energy at each fault also means that the clarity of the discontinuity recedes along a sequence of faults. JTFA utilizes an excitation signal carefully characterized by time and frequency content to address this challenge [13,76,80]. Applications of JTFA in nuclear power plant cables, high-temperature superconductor cables, and in high-voltage direct-current submarine cables have been explored [75]. Pure FDR is another variation in this field, forming a large body of work. Working in the frequency domain, strain sensing at multiple locations along a cable and greater resolution on strain measurements could be achieved. This is apparent on a comparison of specifications of TDR analysis equipment and FDR analysis equipment (Vector Network Analyzer (VNA)) [81,82]. The sampling rate of the TDR instruments limit resolution and the TDR electronics are more complex and expensive than the equipment needed for FDR [52,76]. The technique of frequency domain coaxial cable strain sensing is described in the next section along with a review of the work conducted in this field and state-of-the-art results.

### 3.2. Coaxial Cable Frequency Domain Reflectometry

Traditionally, FDR was developed for fault diagnosis in electrical systems. A transmitted sine wave is used as a reference against which reflected sine waves, arising from reflectors caused by impedance changes along the cable, are compared and analyzed. A fault, caused by strain or mechanical movement creating a localized change in cable impedance, can be identified and located by analyzing a shift in frequency or phase between the input signal and reflected signal [52,54]. This technique could be susceptible to a high error rate due to the sensitivity of phase to noise [76]. Interferometric techniques, first developed for fiber optic cable distributed sensing, can be applied to coaxial cables, utilizing a different part of the electromagnetic spectrum [15,21]. These methods use artificially created partial reflectors to track changes in the cable and infer environmental conditions such as strain or temperature. By placing partial reflectors along the cable, the magnitude of the partially reflected signals can be controlled through manufacturing and therefore the SNR can be improved, optimizing the FDR technique for condition sensing.

In this report, coaxial cable frequency domain reflectometry is split into two types of sensors, interferometric sensors and grating-based sensors, as outlined in Figure 1. An introduction is given to each technique, followed by a review of the work in that field.

#### 3.2.1. Interferometric Coaxial Cable Frequency Domain Reflectometry

A Fabry–Perot interferometer is an optical instrument constructed from two parallel, partially reflective surfaces or reflectors as shown in Figure 13 [83]. Reflections from each surface interfere, creating a fringed pattern of maxima (at constructive interference) and minima (at destructive interference), as illustrated in Figure 14. The frequency location of these maxima and minima is determined by the electromagnetic path length difference between the reflection from the first partial reflector and the reflection from the second partial reflector. The technique is not constrained to optical frequencies but can work across the electromagnetic spectrum. A Fabry–Perot interferometer is created on a coaxial cable by making a pair of partial reflectors through a localized change in impedance from either a variation in geometry, as illustrated in Figure 13, or material property, such as permittivity or permeability. The theory of using a coaxial cable Fabry–Perot interferometer (CCFPI) as a strain sensor is well documented [14,16,45,46] and outlined here.

The interferogram is created by the superposition of the two reflected waves $U_{1}$ and $U_{2}$, described as follows [46]:(3)$$U_{1}=\Gamma \left(f\right)e^{-\alpha z}cos\left(2\pi ft\right),$$(4)$$U_{2}=\Gamma \left(f\right)e^{-\alpha z}cos\left[2\pi f\left(t+\tau \right)\right],$$(5)$$\tau =\frac{2d\sqrt{\varepsilon_{r}}}{c}.$$

The addition of $U_{1}\;$and $U_{2}$ creates an interference pattern of the following form, as illustrated in Figure 14:(6)$$U=2\Gamma \left(f\right)e^{-\alpha z}cos\left(2\pi f\tau \right)cos\left[2\pi f\left(t+\tau \right)\right],$$
where Г is the reflection coefficient of the partial reflectors, *f* is the frequency of the electromagnetic wave, α is the attenuation of the cable, z is the axial cable direction, t is the arrival time at the first partial reflector, τ is the time delay between the two reflected waves, $\varepsilon_{r}$ is the relative permittivity of the dielectric in the coaxial cable, d is the distance between the two partial reflectors, and *c* is the speed of light.

The fundamental frequency is given by (1/τ) and is dependent on the spacing of the two partial reflectors, as shown below:(7)$$fundamental\;frequency=\frac{c}{2d\sqrt{\varepsilon_{r}}},$$
where *c* is the speed of light, *d* is the distance between the two partial reflectors, and $\varepsilon_{r}$ is the relative permittivity of the dielectric in the coaxial cable.

With increased strain, d will increase through elongation and $\varepsilon_{r}$ will vary due to the photoelastic effect of dielectrics [46]. This will change the fundamental frequency, shifting the interferogram pattern. The frequency location of key features such as the minima or maxima is tracked to infer the change in strain.

The principle of using a CCFPI in a strain sensing application was reported in [46]. Two partial reflectors, with 70 mm spacing, were manufactured onto an RG58 coaxial cable by a hole-drilling method. The reflection spectrum was captured by a VNA, and the frequency of the dips of the interferogram was tracked. As strain was applied, the distance between the two partial reflectors changed and so the resonant frequency changed. A total strain of ~34,000 µε was applied to the cable in 18 steps of 2000 microstrains (0.2%). The frequency demonstrated a quasi-linear response to strain with a relationship of approximately −3.3 kHz/µε, suggesting that a CCFPI had sufficient sensitivity for use as a strain sensor and demonstrating the superior maximum strain capability over fiber optics. The results are shown in Figure 15a. The spatial resolution of the CCFPI is determined by the distance between the partial reflectors, which is 70 mm in this study.

Cheng [14] also studied the application of CCFPI as a strain sensor. The coaxial cable type used in this testing was not stated, but the applied strain was between 0 and 1000 microstrains over 14 steps, and the tracked frequency minima was around 1.91 GHz. The data presented in [46] demonstrate a decrease in resonant frequency as strain increases, implying that the physical elongation with the applied axial strain dominated the response, as illustrated in Figure 15a. Conversely, data presented in [14] demonstrated an increase in resonant frequency with increasing strain, as shown in Figure 15b. The reasons for the conflicting reported results could be due to the differing responses of the cable dielectric permittivity to strain. The variation in the results highlights that the response to strain is complex, and greater research is needed in order to achieve a single-valued, repeatable response to strain that will be required for a successful manifestation of a CCFPI as a distributed strain sensor in industrial applications.

Through different topological configurations of the coaxial cable, the potential of using RG58 as a torsion sensor based on cascaded CCFPIs (Figure 16) [84] and beam shape sensors through strain sensing was demonstrated [85]. For both of these references, testing occurred in the GHz region [14].

A large strain-tolerated Fabry–Perot interferometer smart steel strand has been developed [45], making a CCFPI on SF047 coaxial cable and embedding it into glass fiber-reinforced polymer (GFRP) before replacing the core wire of a steel strand with this GRFP-CCFPI, as depicted in Figure 17. The intention was that this sensor could replace a steel strand in a structural component such as a bridge cable or anchor rod for continuous health monitoring.

The bare SF047 CCFPI demonstrated a measuring range up to 140,000 µε. When embedded in GRFP, the dynamic range was shown to be 16,000 µε. The partial reflectors for this work were created by crimping metal ferrules onto the SF047 cable at two specific locations 200 mm apart. A resonant frequency near 3 GHz was tracked during the strain tests and a strain sensitivity of −3.7 kHz/µε was recorded. This work concluded that the GRFP-CCFPI could replace a previously developed optical fiber sensor-based smart steel strand, offering a much larger measurement range as it can withstand a greater maximum strain. The GRFP-CCFPI sensor response to strain also demonstrated good sensitivity and linearity. Further work in this field was reported in 2024 [87], where a coaxial cable Bragg grating (CCBG) structure was integrated with GFRP into a steel strand. SF047 cable with a diameter of 1.19 mm was used to create a sensing structure comprising 41 discontinuities spaced at 20 mm intervals (total sensing section 800 mm). The increased number of partial reflectors, created by crimping metal ferrules to generate an impedance change, ensured that the SNR was sufficient. The CCBG was then embedded within GRFP before being incorporated into a steel strand. As well as testing to ensure that the CCBG did not compromise the mechanical strength of the GFRP or the steel strand, the strain sensing performance was compared with linear variable differential transformer measurements. The results suggested the CCBG embedded in GFRP could measure strains up to the ultimate limit of GFRP, 20,000 µε. Similar work looking at embedding FBG could only monitor strains up to 6000 µε. The embedded CCBG in GFRP in the steel strand only slightly reduced the mechanical properties of the steel strand, with the tensile strength at 87.9% and elastic modulus at 88.7%. The CCBG structure was reported to demonstrate strain resolution of at least 100 µε and a range of 150,000 µε.

A series of partial reflectors could be manufactured along the entire length of a coaxial cable. Any two consecutive partial reflectors create a Fabry–Perot interferometer (FPI), as illustrated in Figure 18.

In this way, a coaxial cable could become a distributed sensor. Cascading a series of FPIs is possible due to the way in which the VNA captures the S11 reflection spectrum [23]. The entire frequency information, including magnitude and phase, is recorded, which enables the interferogram of each FPI to be uniquely associated with a physical location along the cable [14]. The value of this feature is apparent considering a CCFPI in a monitoring application; this means that the part of the structure experiencing a significant strain event can be identified, and remedial effort can be focused on the area of concern, optimizing efficiency of any maintenance work required.

Several signal processing methods are cited to analyze the CCFPI interferogram signal. The technique that appears to be employed in a lot of the references cited in this paper uses a form of JTFA [14,23,46]. The method of JTFA used in the CCFPI analysis is detailed in the following signal processing steps.

A VNA is used to send a frequency swept signal and capture the S11 signal amplitude and phase information from the entire CCFPI cable length.In the time domain, the gating function on the VNA is applied to a single pair of partial reflectors, creating an individual CCFPI. The time span of this gate is designed to encompass the two reflections but must be wide enough to avoid breach of the minimum time gate span rules specified in the VNA documentation, which is dependent on the bandwidth of the swept frequency signal. Applying the time gate to a single CCFPI sufficiently isolates it from the rest of the cable features for analysis. With reference to Figure 18, the red dashed line indicates the principle of time domain gating, isolating a single CCFPI.Fourier transform is used to convert the gated time domain signal of a single CCFPI into the frequency domain; this feature is often built in to the VNA functionality. In the frequency domain, the interferogram of the isolated CCFPI under analysis will be apparent, such as the interferogram given in Figure 19.This action is repeated for all pairs of reflectors along the cable, sliding the time domain gating window along the cable, isolating each individual CCPFI in turn for analysis. This is a method of short-form Fourier transform and enables each CCFPI to be monitored and, crucially, the physical location of each CCFPI on the cable is known, through the time gate, allowing parameters such as strain along the cable to be monitored and mapped along the cable length.

Using this technique, different features of the interferogram can be tracked, although the main feature to be tracked appears to be the resonant frequencies or interferogram minima, illustrated below in Figure 19. In their development of a CCFPI for sensing applications, Huang et al. report tracking the dips in the interferogram and also observe that the Q-factor of the dips decreased as strain increased, which could indicate an increase in the propagation loss between the two reflectors [46]. A similar approach was taken in the development of a smart steel strand with built-in CCFPI, illustrated below in Figure 19. The sharpness of the dips compared to the peaks could provide greater resolution.

A different joint time–frequency analysis method, time–frequency domain reflectometry (TFDR), was proposed by Song et al. [76]. This was presented as a method of overcoming the challenges of TDR and FDR on unmodified coaxial cables. The argument was that for accurate TDR, a sharp rise in the time pulse is required, but this incurs distortions. Conversely, FDR is very sensitive to noise and so carries a high error rate. By performing the analysis jointly in the time and the frequency domains, this method aims to overcome the challenges incurred when analyzing in a single domain. This is linked to the uncertainty principle, giving rise to the trade-off between the time duration and the frequency bandwidth of the same signal. In order to analyze both the time and frequency domains simultaneously, the reference signal launched down the cable was chosen to be a Gaussian waveform envelope (frequency domain) and a chirp signal (time domain), as illustrated in Figure 20.

The reflected signal is then cross-correlated with the reference signal to reveal the location of faults along the cable. This method was shown to offer improved fault location accuracy on unmodified coaxial cables, but the inherent attenuation of the coaxial cable resulted in decreasing accuracy and sensitivity with increased fault distance [76]. This reference does not document the sensitivity of this technique to lower strain events and only considers identifying the location of significant cable damage (absence of outer conductor material) on an otherwise unmodified coaxial cable.

A novel signal processing technique of using a sliding time gate and cross-correlating spectral data is well described in [88], where a series of cascaded partial reflectors formed the sensing region of a coaxial cable. Time domain plots of the technique demonstrate the limited measurement resolution and clarify the need to perform frequency domain analysis. The correlation technique is interesting in this context and is combined with a structure comprising a series of 100 randomly spaced weakly reflecting holes to create a 2 m long strain sensor that could have applications to CCFPI.

The spatial resolution that can be achieved with CCFPI is dependent on the spacing between the two partial reflectors constituting the FPI. There is an inherent trade-off between the frequency bandwidth and spacing, with a higher bandwidth required to achieve greater spatial resolution. The bandwidth required to measure the interferogram varies with the spacing of the partial reflectors. The partial reflectors cannot be resolved unless they are greater than *x* distance apart [14]:(8)$$x\;must\;be\;\geq \frac{c}{\left(\omega_{max}-\omega_{min}\right)*2*\sqrt{\varepsilon_{r}}}\;,$$
where *x* is the distance between the two partial reflectors making up the FPI, $\omega_{max}$ is the maximum frequency and $\omega_{min}$ is the minimum frequency, $\varepsilon_{r}$ is the permittivity of the coaxial cable dielectric, and *c* is the speed of light.

Whilst the focus of this report is on coaxial cable distributed strain sensing, there are several derivatives of this technique providing a single discreet measurement, or tailored to track a parameter different to strain, which are worthy of note.
On creating a pair of highly reflective reflectors in a coaxial structure, a Fabry–Perot resonator (CCFPR) is constructed. Whilst only a single measurement can be made from this device, multiplexing is not possible as insufficient energy passes the reflectors and the multiple round trips of the energy within the cavity increase the Q-factor of the device, increasing the measurement resolution. The second reflector forming the resonant cavity can be placed beyond the open end of the coaxial structure, thereby forming ‘open-ended coaxial probes’ which are widely available for measuring material properties in the microwave range of the electromagnetic spectrum [89,90,91,92,93,94,95,96]. By adapting this arrangement, it was shown that the second reflector in the CCFPR could be formed by a metal plate positioned beyond the end of the coaxial structure. The lateral position of the plate could be measured to resolutions of the order of 1 nm, comparable with the resolutions of analogous fiber optic techniques [97,98]. A diagram of this open-ended hollow coaxial cable resonator (OE-HCCR) is shown in Figure 21.

As a further extension of this work, the coaxial resonator was adapted to measure vibrations and impacts and, combined with machine learning techniques, the cause of the impact could be identified [99]. A diagram of this OE-HCCR impact sensor is shown in Figure 22.

A humidity sensor has also been developed based on this technique, harnessing the sensitivity of the device to detect the moisture content of exhaled air. The potential applications include chemical sensing and the analysis of gaseous contents [100].

A similar OE-HCCR construction was devised as a strain sensor for nanoscale precision [101]. The premise of the operation was to track a frequency shift occurring from a change in the reflection coefficient of the open end of the OE-HCCR. The reflection coefficient was dependent on the signature from a gap and a flange. With varying strain, the gap width changed, altering the capacitance and leading to a reflection coefficient change. Sensitivities of 2.5 GHz/mm were achieved. An application to measure shrinkage strain during the mortar drying process was demonstrated.
A displacement sensor based on a hollow coaxial cable Fabry–Perot resonator (HCC-FPR) was developed and tested in 2018 [102]. A solid stainless-steel inner conductor (6 mm diameter) and a tubular stainless-steel outer conductor (14 mm diameter) form the basis of an air-dielectric coaxial structure. A single pair of highly reflective partial reflectors exist, with one on the moveable handgrip, forming a resonant cavity. When the handgrip is moved, the reflector moves, changing the length of the resonant cavity and causing a shift in frequency. The device, shown in Figure 23, could measure displacement to a resolution of 10 µm.


Further developments on the HCC-FPR proved the principle of this device as a strain sensor for high-temperature environments, up to 1000 °C. The movable hand grip was replaced with weld points (limit disks) to attach the device to a test steel plate. Now, as the steel plate expands with temperature, the cavity length of the FPR changes, shifting the resonant frequency, from which strain can be deduced. The nested arrangement of the coaxial structure is intended to remove the effects of the thermal expansion of the sensor itself [103]. The principle of the operation of this device is illustrated in Figure 24, along with photographs of the tested sensor.



Adapting a coaxial cable into a CCFPR for temperature measurements was described in 2017 [104]. In contrast to utilizing partial reflectors in distributed sensing using FPI arrangements along the entire length of the cable, CCFPRs use a single pair of highly reflective points, in this case constructed by filling drilled cavities in a coaxial cable with copper powder to form a short circuit (Figure 25). The high reflectivity results in increased measurement resolution [105] but limits the number of sensing points. This investigation noted the adaptability of the technique to strain sensing.



The application of an HCC-FPR as a liquid-level sensor was researched in [106]. In this configuration, illustrated in Figure 26, the liquid forms the second reflector creating the resonant cavity, and hence the level of the liquid determines the frequencies of resonance. The sensor could measure liquid levels over a ~20 cm range to resolutions in the order of micrometers.



Characterizing liquids through the measurement of their dielectric properties is important for a range of functions such as food processing, biological analysis, and the design of microwave communication systems. A sensor based on a CCFPI construction was developed for this application in 2017 [107]. A bespoke coaxial structure was manufactured using a stainless-steel tube and wire as the outer and inner conductors and ceramic (Al_2_O_3_) as the dielectric. Two Teflon disks formed the partial reflectors either side of a cavity formed by the omission of the ceramic dielectric for a ~10 cm proportion of the structure. The construction of the device is shown in Figure 27. Different fluids were pumped through this cavity, in turn, and the resultant interferogram was shown to be dependent on the dielectric properties of the different fluids.



A metal–ceramic coaxial cable design was proposed for high-temperature monitoring using the Fabry–Perot interferometric technique [108]. The high-temperature properties of the ceramic replace the temperature-limited conventional polymer dielectrics. Successful operation was reported between 200 °C and 500 °C, although thermal stability over longer time frames at these temperatures was yet to be explored. Two different partial reflector designs were tested: full-circle air gaps and half-circle air gaps, as shown in Figure 28.



The use of CCFPI for temperature monitoring is also described in [109], where copper crimp rings are compressed onto a conventional coaxial cable to form an FPI (~10 cm) long. The objective of this research was to design a sensor capable of monitoring temperatures’ downhole in order to indicate leakages in CO_2_ storage.


#### 3.2.2. Grating-Based Coaxial Cable Frequency Domain Reflectometry

Fiber Bragg gratings are well established in the field of fiber optic sensing, as discussed in Section 2.2.2 of this paper. There are wide-ranging examples of optical FBGs as sensors [110,111,112,113,114]. However, the inherent weakness of the fiber optic material limits its application to lower strain events. To realize a wider range of measurable strain and monitor structures up to failure, a more robust FBG ‘carrier’ would be necessary. The application of an FBG structure on a coaxial cable achieves this [15,21]. Whereas a CCFPI is constructed from a pair of partial reflectors, a coaxial cable Bragg grating consists of a group of equally spaced partial reflectors (as shown in Figure 29), all forming an interferogram that tracks with changing environmental conditions.

The proposed benefit of CCBG over CCFPI is the higher Q-factor of the interferogram, which should offer increased measurement resolution [104,105,115]. The CCBG interferogram clearly shows sharper maxima, indicating a higher Q-factor, over the broader peaks of the CCFPI interferogram, as illustrated in Figure 30.

Wei, Wu, and Huang et al. reported on fabricating a CCBG in 2011 [21]. A series of 46 holes, 2.5 cm apart, were drilled onto an RG58 cable, penetrating the dielectric layer, to create a series of partial reflectors (analogous to the changes in refractive index manufactured into FBGs). The holes present a localized change in impedance, which gives rise to a reflection as the discontinuity is encountered, whilst still allowing for most of the signal to propagate along the cable. A VNA was used to measure the S11 return loss in the frequency domain configured to capture the first resonance peak. A load frame was used to apply a total axial strain of 20,000 µε in 16 steps. The frequency of the first resonant peak showed a linear response to the applied axial strain. The maximum strain applied (20 mε) was significantly greater than a typical fiber optic could withstand (about 4 mε), proving the principle that a coaxial cable is a more robust alternative to fiber optic technology for strain sensing in challenging environmental conditions [21].

A similar investigation was conducted using an FBG of 41 discontinuities (holes) with a gating period of 25 mm to measure a 1 m length of RG58 cable subjected to axial strain [15]. This CCBG again demonstrated a large dynamic strain of around 5% with a linear response to strain and a resolution of 100 µε. From the results, it was observed that the Q-factor decreased as the strain increased. This was explained due to the change in impedance mismatch at the site of the Bragg gratings as the cable dimensions change with increased strain. In this paper, a cross-correlation method was applied between the interferogram before and after applied strain to detect the small shift in resonant frequency.

A disadvantage of a Bragg grating is the reduced spatial resolution that can be achieved [14,45]. Monitoring a Bragg grating of 46 periods of 2.5 cm distance covers a total distance of 1.125 m. Information on the strain deduced from tracking the interferogram of this Bragg grating could therefore only be assigned to a location covering a 1.125 m span.

At the 2012 workshop on civil structure health monitoring, the results on the development of a CCBG strain sensor were presented. A variety of cable types were tested, and it was demonstrated that the dynamic range of the sensors was up to 7% (70,000 µε), measuring strain to a resolution of 100 µε. The signal analysis techniques included a cross-correlation method to improve the sensitivity of the device. Furthermore, a positive feedback oscillator analog system was developed and shown to increase the Q-factor of the data from the device by ~3500 times. This improved the strain resolution by almost a factor of 10, from 100 µε to 11.4 µε [115].

Further work and details of the positive feedback oscillator to increase the sensitivity to strain were explored and reported in 2017 [116]. An RG58 cable formed the basis of the CCBG sensor under test. A series of ~8 partial reflectors were created by milling out the outer conductor and dielectric in discreet locations ~11 cm apart. This manufacturing method is shown in Figure 31.

The emphasis of this research was to make a cost-effective geodetic strain sensor. The proposed solution was to use a pair of CCBGs: one detecting strain, the other detecting environmental noise. Integrating the two signals in a mixer allowed for measurements to be taken at a lower frequency, which reduced the complexity and cost of the electronic components needed for signal propagation and analysis. A cost-effective portable spectrum analyzer could therefore be used for data processing, eliminating the need for an expensive VNA.

Recent work in the field of CCBG (2023) investigated designing and constructing a bespoke coaxial cable, with a regularly undulating dielectric layer forming the grating feature, to operate as a strain sensor via the CCBG technique. A thermoplastic polymer was chosen as the dielectric layer, offering greater elongation and tensile strength over conventional polyethylene dielectric, to increase the range over which a CCBG strain sensor could operate before undergoing plastic deformation and become unusable. The grating feature was constructed from a regularly fluctuating cross-section of the dielectric, as illustrated in Figure 32. This idea built on work conducted in 2013 which proposed creating partial reflectors on a CCBG structure by modifying the cross-section of the coaxial cable instead of hole drilling [117]. The proposed benefit of this approach was to eliminate sharp impedance discontinuities formed by crimping or drilling a coaxial cable, which introduces a mechanical weakness [118]. HFSS modeling was employed to determine the optimum shapes of the dielectric before a variety of dielectric designs were 3D-printed, and the bespoke cables were constructed for the test. The most sensitive CCBG had the most extreme variation in the dielectric cross-section and demonstrated a resonant frequency shift of ~3.075 kHz/µε [118].

Through the inclusion of a series of weak partial reflectors along a coaxial cable, the entire structure can be converted into a sensor. Recent work in 2024 [88] demonstrated the promising performance of a coaxial cable sensor, made from a series of 100 randomly spaced holes creating partial reflectors. This forms a type of irregular Bragg grating. The random arrangement generated a multibeam interference pattern. Combined with a signal analysis technique utilizing a sliding time gate and cross-correlation techniques, strain data were inferred across a 2 m section of a 15 m long MIL-C-17 coaxial cable. Strains up to 18,000 µε were measured. The resolution of the strain measurement was shown to be dependent on the spatial resolution (spatial width of the time gate). Increasing the time gate width (inclusion of more partial reflectors in the measurement) improved the SNR and measurement resolution. For higher-spatial-resolution measurements, where the time gate was narrower, fewer partial reflectors were included in the measurement, lowering the SNR and measurement resolution. The trade-off between reflector size, signal-to-noise ratio, and successful transmission distance was noted. Larger reflectors improve the SNR but reduce the total distance over which signals can propagate along the device.

A summary of the key results from published works on the development of coaxial cable strain/displacement sensing is given in Table 4.

## 4. Future Research Challenges in Coaxial Cable Strain Sensing

An extensive range of exciting developments in the field of CCFPI strain sensing have been investigated under laboratory-scale tests designed for practical applications. However, CCFPI is not yet a standard, industrial SHM tool, and further work remains to promote this technology to a workable product. Understanding the longevity and stability of CCFPI-based devices is necessary before their deployment in field applications [16]. Fiber optic technologies have long since provided an industrial solution to a large range of SHM requirements. CCFPI needs to prove that it can provide additional value to complement the DOFS before it can become a viable, commercial alternative. The potential advantages of utilizing coaxial cables as distributed strains sensors, complementing or replacing incumbent fiber optic technologies, are several-fold.

Higher strain events can be monitored due to the strength of the materials of coaxial cable construction [9,46].Coaxial cables can withstand greater exposure to challenging, harsh environments [21,45].The robust coaxial cable structure means that they can withstand installation procedures which are too severe for delicate fiber optics [10].The construction and manufacturing techniques and signal excitation and analysis methods are all simplified compared to fiber optic technology, offering the potential for a more economical alternative [2,47].

However, from the reviewed literature, it is apparent that for the concept to progress to a usable technology, there are several research challenges remaining, including the following:A CCFPI response to strain is dependent on several factors, not just the physical distance between the two partial reflectors comprising the FPI. The response of the dielectric permittivity to strain has a significant influence on the performance of a coaxial cable as a strain sensor using FPI features. Characterizing the dielectric permittivity with strain is an area for further work. This would enable the likely response of different coaxial cables as strain sensors, using FPI features, to be understood and predicted.CCFPI-distributed sensing measurements can be a result of the convolution of strain and temperature effects on the device. Both environmental factors influence the interferogram. Suggestions on how this challenge may be addressed, and the effects of the proposed decoupling methods, are summarized below and in Table 5:
Inference is made that the convolution of strain and temperature effects can be minimized by selecting dielectric materials that show preferential sensitivity to strain or temperature, depending on which condition is to be monitored [14]. There is a further reference to this technique in [7] where instead of using a commercially available coaxial cable with a Teflon or polyethylene dielectric, a coaxial cable with a low-stiffness rubber dielectric was designed in an attempt to increase the sensitivity of the cable to strain.Another approach taken has been to use a reference CCFPI alongside the CCFPI monitoring the strain to act as temperature compensation [14]. This is similar to the well-established practice of utilizing conventional electrical resistance strain sensors in a bridge arrangement to provide temperature compensation [119].Depending on the application and CCFPI design, temperature variations could be accounted for by presentation as an error on the strain reading, within reason.Lessons could be learned from fiber optic techniques and are worth future assessment and investigation [40,120,121,122,123,124,125].Signal analysis methods to monitor different characteristics of the interferogram might hold the key to inferring temperature effects on strain measurements. The latest developments in machine learning could more rapidly classify interferogram changes due to temperature and changes due to strain, deconvolving the environmental effects.Recent work on FBG sensors has applied machine learning techniques to discriminate between strain and temperature variations [126]. It would be interesting to pursue the role machine learning could play in advanced signal processing techniques for CCFPI.Significant experimental works need to be conducted to test the CCFPI operation when subjected to combinations of environmental parameters and assess operational limits under conditions of high humidity, high pressure, and high temperature. The impact of cumulative and combined environment effects needs to be understood to make sure CCFPI is reliable and robust for real-world applications.In the reviewed work, there have been two main methods for creating the partial reflectors in the FPI arrangement: crimps/localized deformation and hole drilling [15,21,45,46]. For the measurement of strain, it was cited that localized deformations with metal ferrules ensure that cable strength is retained, whereas hole drilling would create undesirable weaknesses [115]. One purpose of the development of the undulating cross-section CCBG was to propose a design that could be realistically fabricated [118]. The practicality and engineering challenges of manufacturing partial reflectors on a large scale, integrated with coaxial cable manufacturing processes, should not be overlooked to realize the full commercial potential of this technology. This is another aspect worthy of further work.Optimal signal analysis techniques would further enhance the CCFPI technology for commercial success. Some techniques reviewed under this paper, but not necessarily applied yet to CCFPI devices, include the following:
JTFA method applied to date to unmodified cables [76].Increased sensitivity of CCBG device achieved through cross-correlation analysis and a feedback oscillator [115].Deeper investigation into key features of the interferogram to track would be interesting, such as amplitude, Q-factor, as well as the position of frequency maxima and minima. The inclusion of machine learning methods into the signal analysis would be a novel approach to extracting greater information from the interferogram data. The objectives of more sophisticated signal analysis techniques would include the following:
Increased sensitivity;Improvement in the SNR;Rapid analysis of multiple FPIs for real-time monitoring;Potential to de-convolve strain and temperature.

## 5. Conclusions

Sensors for structural health monitoring provide vital information to ensure the safety and economic viability of engineering assets. Novel technologies in this field are of interest to keep pace with engineering developments, as well as improve current sensing devices. This paper introduced the concept and principles of distributed sensing for SHM by providing a brief overview and history of the development of optical fiber sensing. Time domain reflectometry was the first technique utilized to infer parameters such as strain from fiber optics, with frequency domain reflectometry arising to address some of the limitations of TDR, such as measurement resolution. A parallel development journey is echoed in the emergence of coaxial cable distributed sensing techniques. Initial work on TDR on coaxial cables demonstrated the principle that such cables could be adapted into distributed sensors able to withstand greater strains than delicate optical fibers. Through the adaptation of interferometric and grating techniques first used on fiber optics, coaxial cables have been transformed into distributed sensors, capable of high measurement resolution up to strains beyond fiber optic limits. Furthermore, the construction and materials of a coaxial structure can be selected and adapted to a wide range of topologies. A standard cable form factor can provide strain information when embedded in a steel rope for anchoring, and a shorter more rigid coaxial structure can form a displacement sensor for high-temperature environments. Coaxial cable frequency domain measurement techniques could therefore offer advantages over distributed optical fiber sensors for the given applications. This paper has documented the state of the art in this promising technology.

Optimizing the design and construction of the features required for CCFPI and CCBG structures is a focus for further research. Some novel manufacturing techniques, reported in this paper, have been devised and tested, and research on these, as well as the underlying physics of the techniques, is necessary to elevate this concept into a successful product, rivaling DOFSs. Enhanced signal analysis techniques could further develop the full potential of coaxial cable frequency domain-based sensors and, as before, inspiration could come from mirroring DOFS techniques and emerging methods.

## Figures and Tables

**Figure 1 sensors-25-00650-f001:**
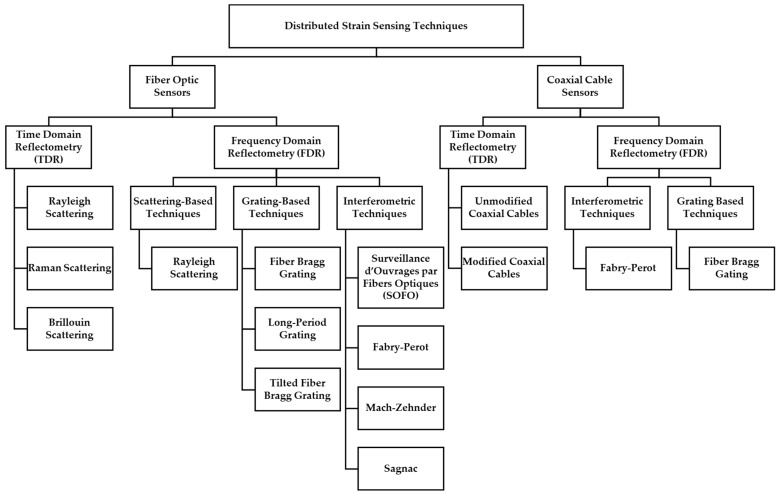
Overview of techniques employed in distributed strain sensing.

**Figure 2 sensors-25-00650-f002:**
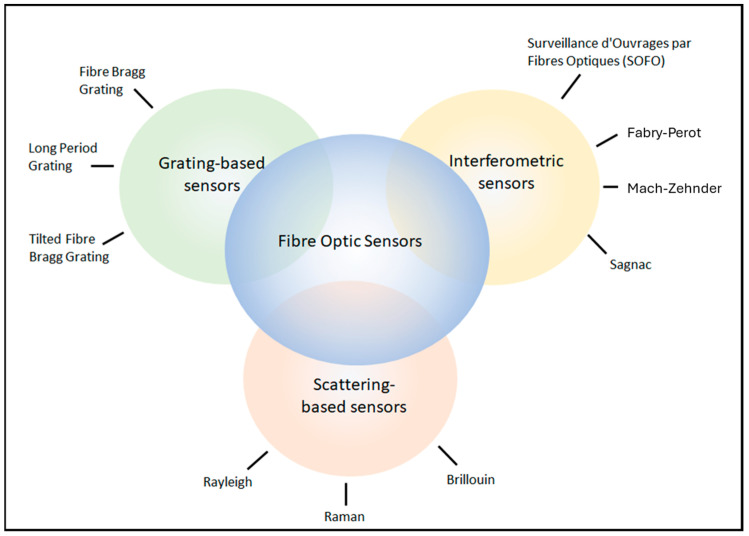
Types of fiber optic sensors (adapted from [10]).

**Figure 3 sensors-25-00650-f003:**
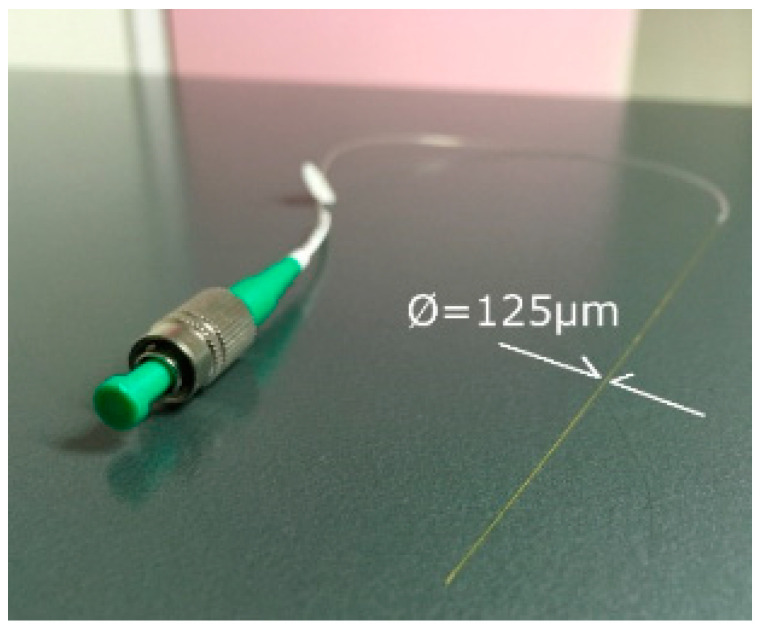
A distributed optical fiber sensor (DOFS) manufacture by LUNA technologies [8].

**Figure 4 sensors-25-00650-f004:**
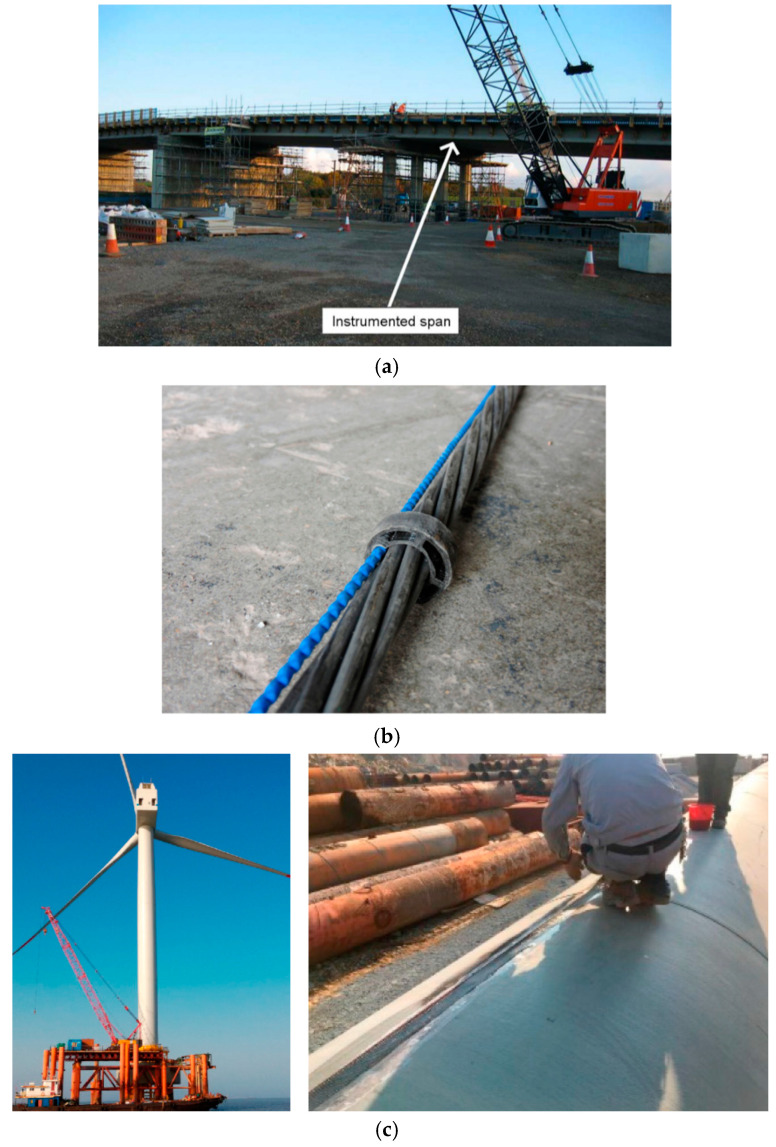
Photographs of distributed optical fiber sensor applications: (**a**) monitoring bridge integrity, (**b**) tracking the stability of ground anchors, and (**c**) measuring strains in a wind turbine pile [8].

**Figure 5 sensors-25-00650-f005:**
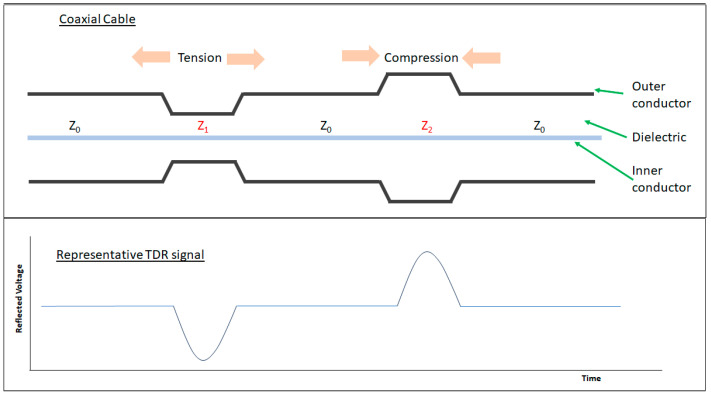
Illustration of the principles of coaxial cable time domain reflectometry (CCTDR) adapted from [48,53], showing a cross-section of a coaxial cable experiencing tension and compression and the associated time domain reflectometry signal.

**Figure 6 sensors-25-00650-f006:**
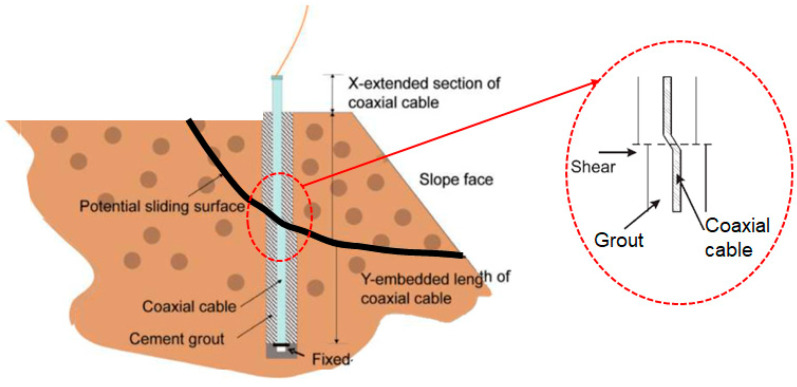
Modeling of shear displacement time domain reflectometry sensing (adapted from [69]).

**Figure 7 sensors-25-00650-f007:**
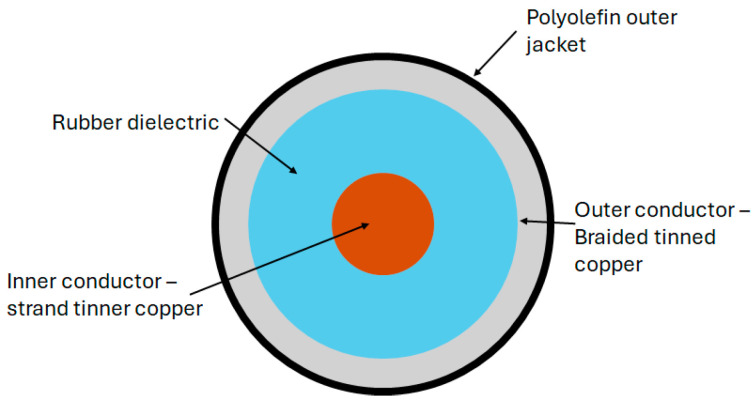
Diagram of a prototype coaxial cable designed for enhanced TDR capability through inclusion of a compliant rubber dielectric.

**Figure 8 sensors-25-00650-f008:**
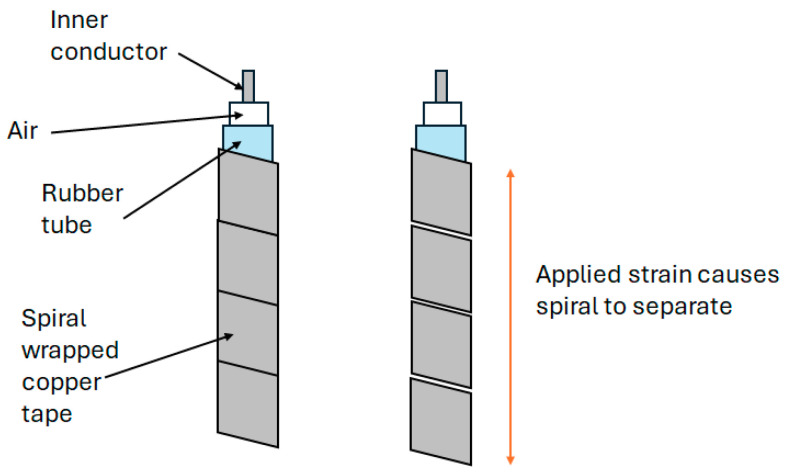
Diagram showing helically wound outer conductor topology.

**Figure 9 sensors-25-00650-f009:**
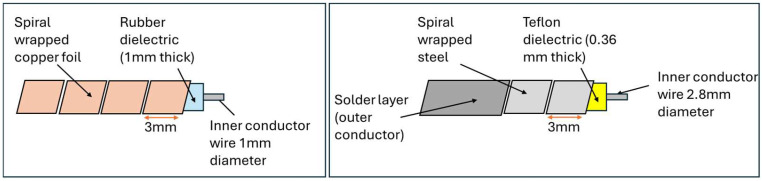
Schematic of a rubber dielectric coaxial cable structure and a Teflon dielectric coaxial cable, both with spiral-wound outer conductors for increased TDR sensitivity [72].

**Figure 10 sensors-25-00650-f010:**
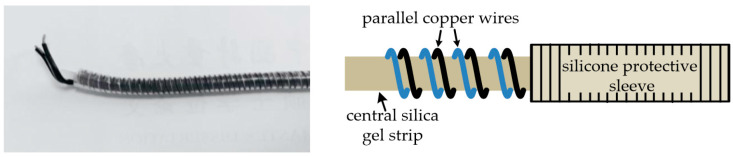
Two parallel conductors helically wound around a central core for TDR tensile deformation monitoring [74].

**Figure 11 sensors-25-00650-f011:**
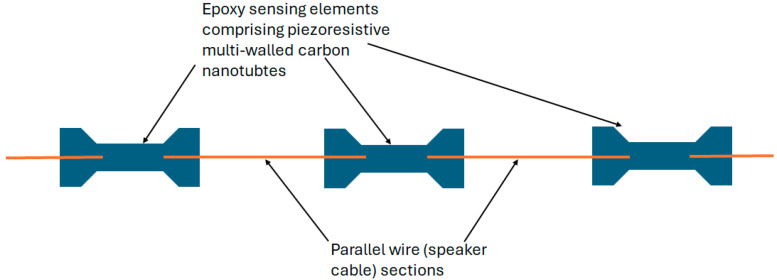
Schematic of novel piezoresistive multi-walled carbon nanotube sensing elements placed in-line with a cable to create a distributed strain sensor [53].

**Figure 12 sensors-25-00650-f012:**
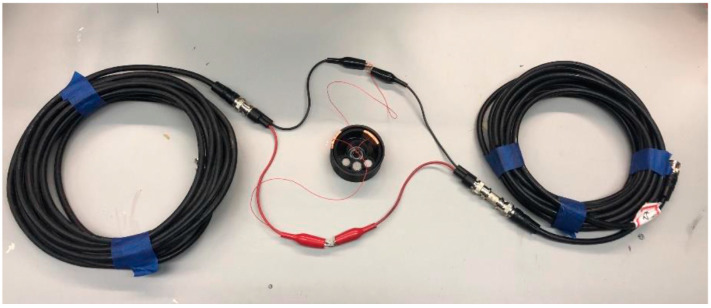
A tilt sensor placed in-line with RG-6 coaxial cables to form distributed sensing [73].

**Figure 13 sensors-25-00650-f013:**
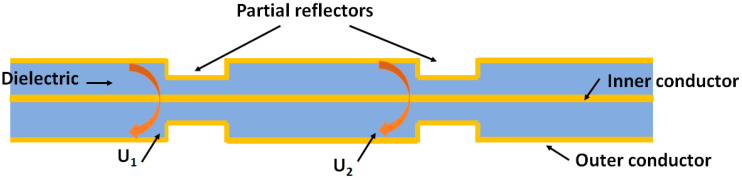
Schematic of coaxial cable Fabry–Perot interferometry.

**Figure 14 sensors-25-00650-f014:**
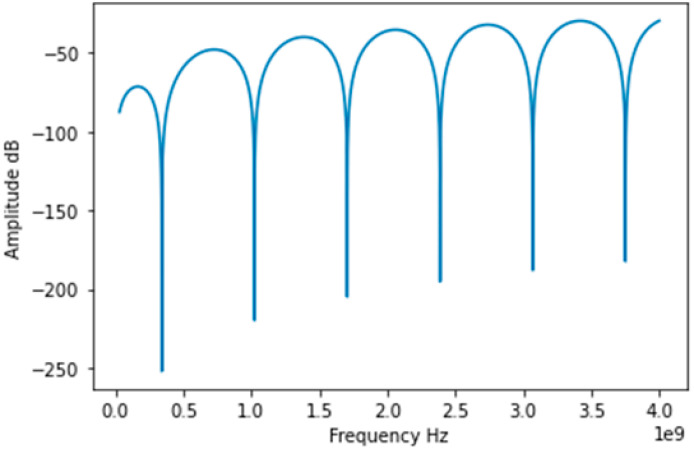
Example interference pattern (interferogram) from a CCFPI.

**Figure 15 sensors-25-00650-f015:**
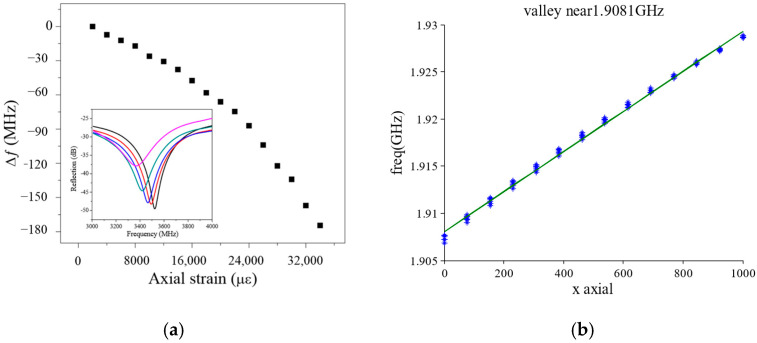
Response of a CCFPI to strain from two different sets of published data showing that the impact of strain on a CCFPI resonant frequency can produce different results. (**a**) Resonant frequency shift as a function of strain with inset showing shift in reflection spectra as strain increases [46]; (**b**) interference resonant frequency shift as a function of applied strain [14].

**Figure 16 sensors-25-00650-f016:**
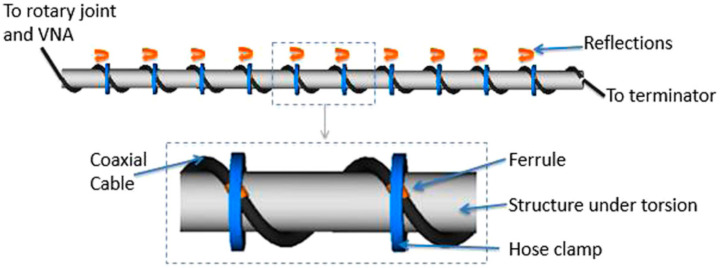
Schematic of using an RG58 CCFPI as a torsion sensor [14].

**Figure 17 sensors-25-00650-f017:**
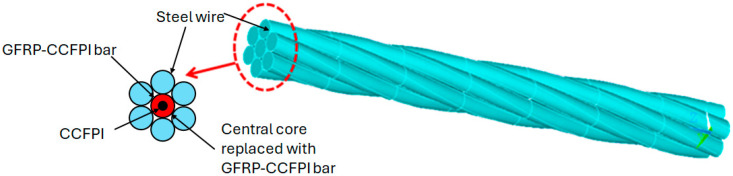
Schematic of the CCFPI SF407 cable embedded in GFRP and replacing the core wire of a steel strand (adapted from [86]).

**Figure 18 sensors-25-00650-f018:**
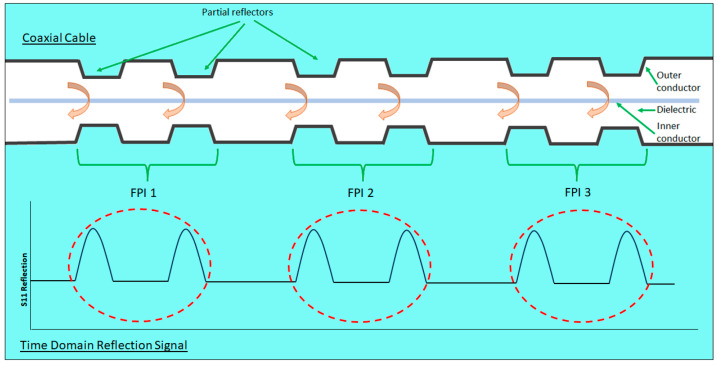
Diagram to illustrate the principles of multiplexing FPIs along entire cable length, forming a distributed sensor.

**Figure 19 sensors-25-00650-f019:**
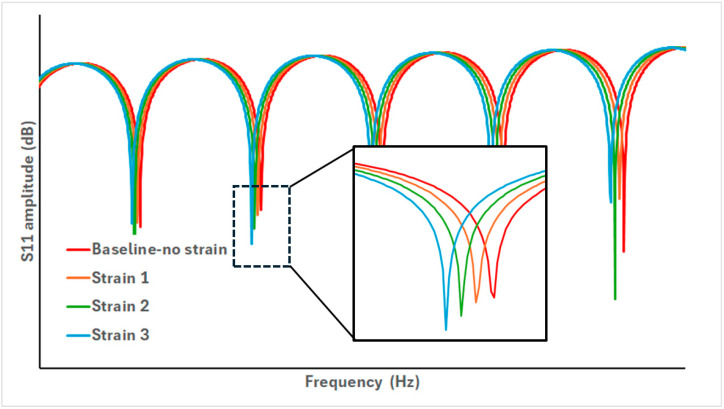
Example interference pattern of a CCFPI sensor varying with applied strain. The zoomed-in section illustrates how strain can be inferred from tracking the frequency of the interferogram minima [45].

**Figure 20 sensors-25-00650-f020:**
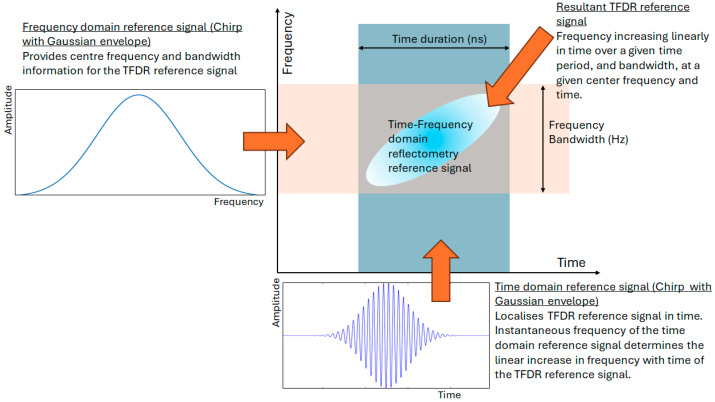
The design of a reference signal for TFDR method (modified from [13,76]).

**Figure 21 sensors-25-00650-f021:**
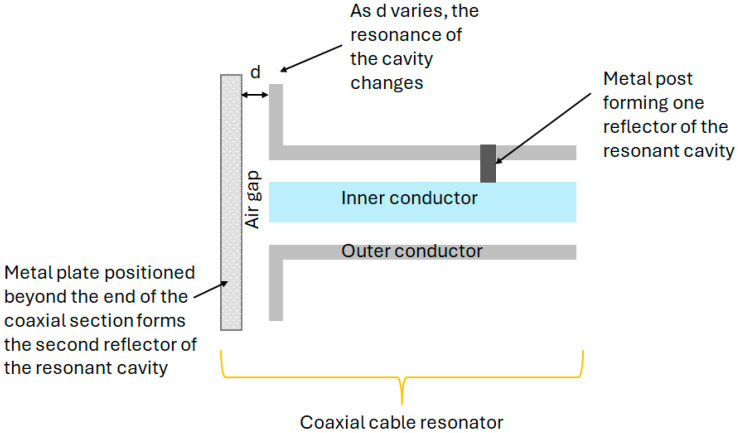
Schematic of the OE-HCCR structure for measuring lateral displacements. The metal post forms the first reflector of the resonant cavity and the metal plate beyond the end of the coaxial structure forms the second reflector of the cavity.

**Figure 22 sensors-25-00650-f022:**
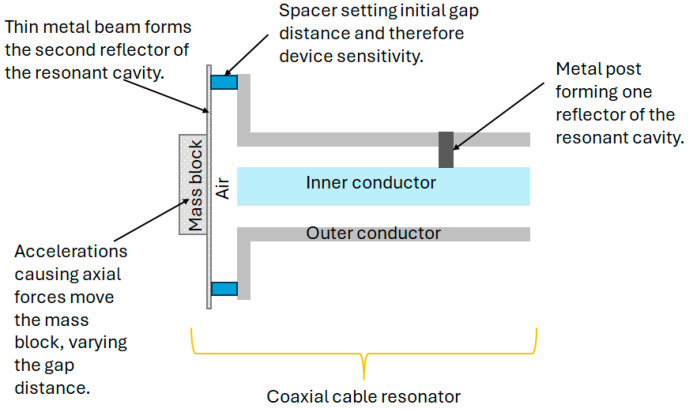
A diagram of an open-ended hollow coaxial cable resonator impact sensor [99].

**Figure 23 sensors-25-00650-f023:**
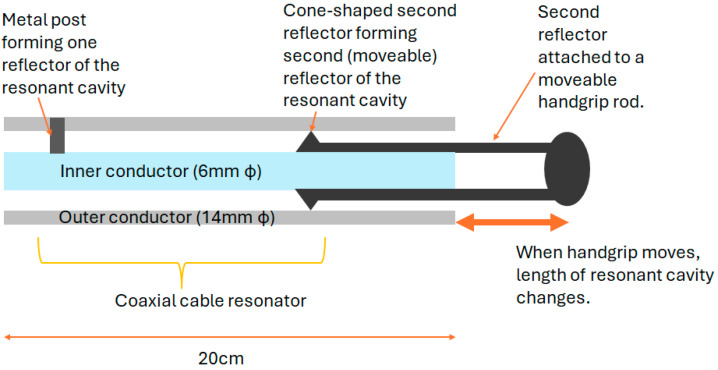
Schematic of the HCC-FPR displacement sensor [102].

**Figure 24 sensors-25-00650-f024:**
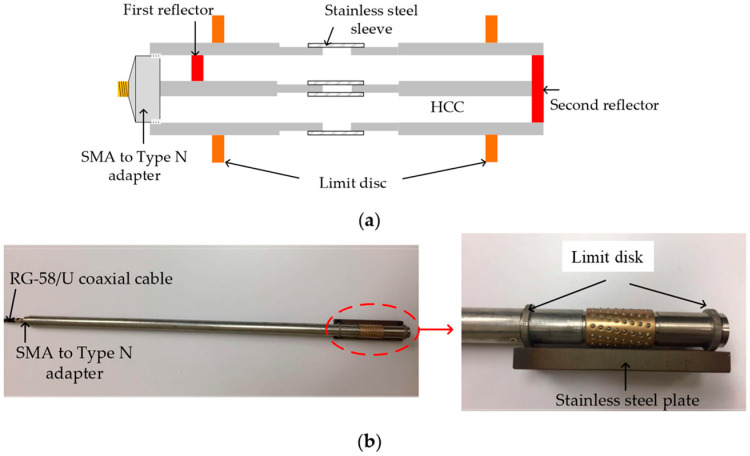
(**a**) Schematic of the HCC-FPR strain sensor. (**b**) Photographs showing the two limit disks welded to the stainless-steel plate for high-temperature strain sensing. The distance between the two limit disks was ~11.5 cm [103].

**Figure 25 sensors-25-00650-f025:**
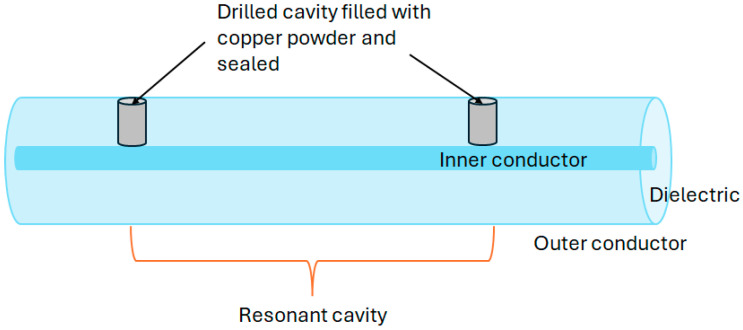
Diagram showing cut-through schematic of the high-quality factor coaxial cable Fabry–Perot resonator [104].

**Figure 26 sensors-25-00650-f026:**
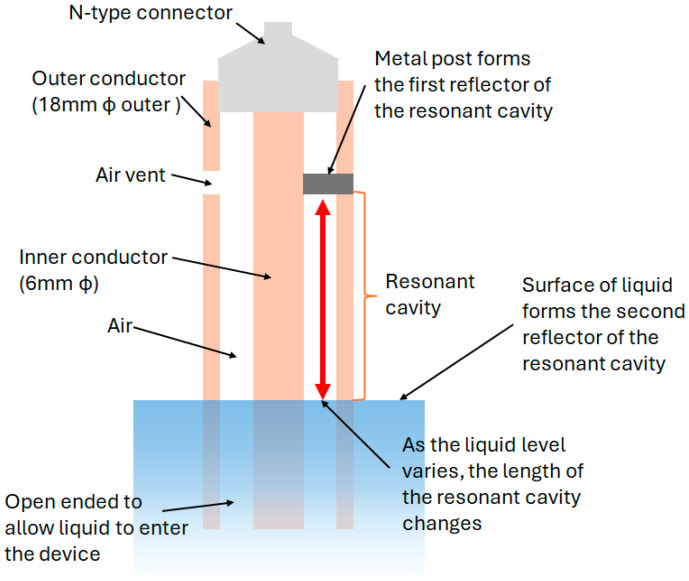
Diagram and of the HCC-FPR device as a liquid-level sensor [106].

**Figure 27 sensors-25-00650-f027:**
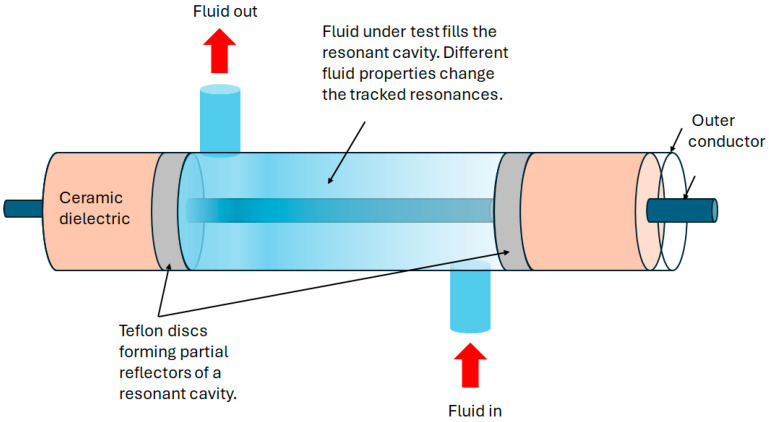
Diagram of the metal–ceramic CCFPI for the measurement of dielectric properties of liquids [107].

**Figure 28 sensors-25-00650-f028:**
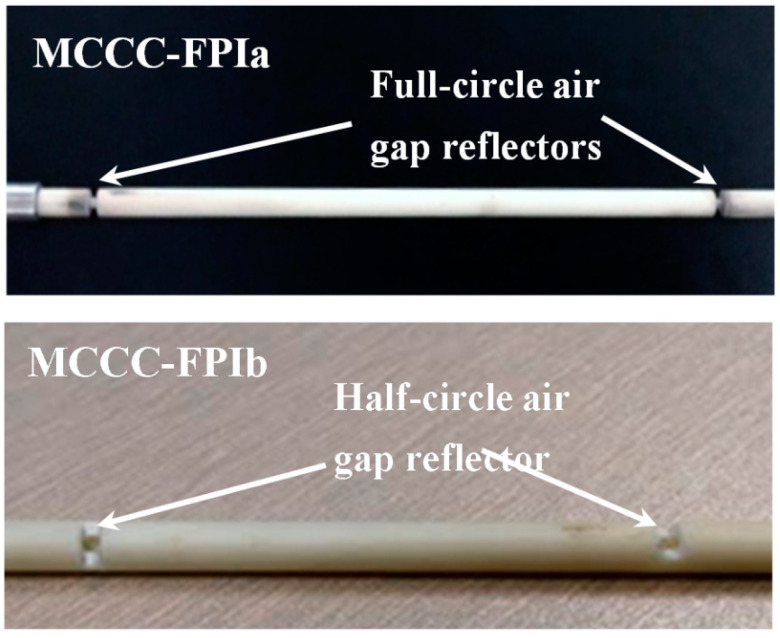
Photograph of the metal–ceramic coaxial cable Fabry–Perot interferometer design with two different configurations for the partial reflectors [108].

**Figure 29 sensors-25-00650-f029:**
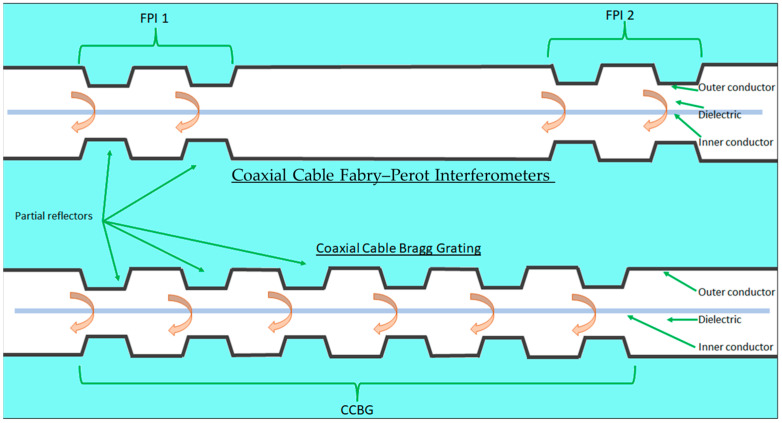
Diagram to illustrate the difference between a CCFPI and CCBG configuration.

**Figure 30 sensors-25-00650-f030:**
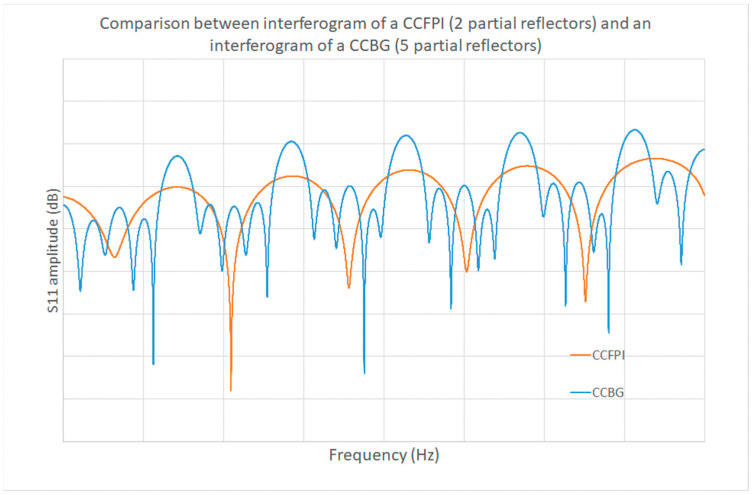
The higher Q-factor of a CCBG over a CCFPI is apparent on studying the interferograms from the respective structures.

**Figure 31 sensors-25-00650-f031:**
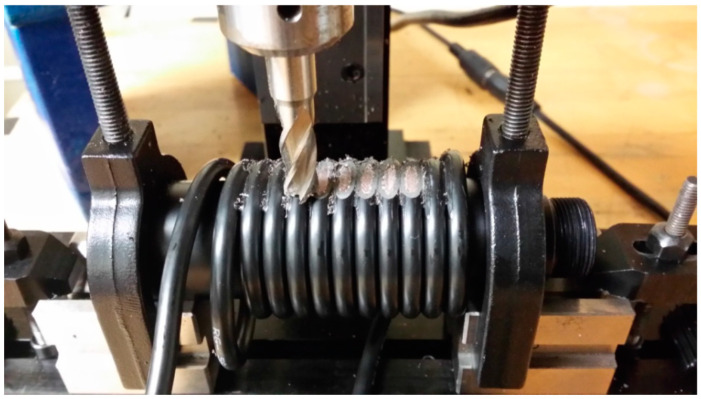
Photograph illustrating the process by which the CCBG was created on RG58 cable [116].

**Figure 32 sensors-25-00650-f032:**
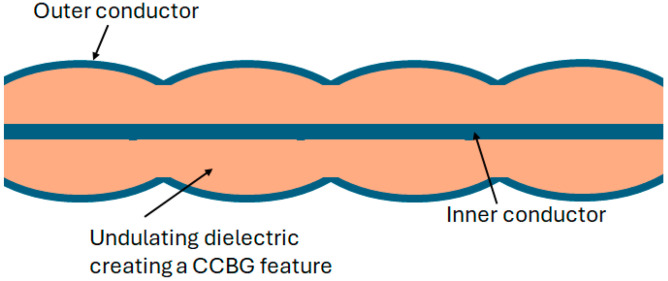
Diagram of bespoke CCBG with regularly undulating dielectric layer forming the grating feature [118].

**Table 1 sensors-25-00650-t001:** A summarized comparison of existing and emerging strain sensing technologies for structural health monitoring.

In Situ Strain Sensor Technology	Basic Principle of Operation, Key Attributes, and Limitations	Quantified Performance Values	Reference
DOFSs (Brillouin or Rayleigh scattering)	Optical time domain reflectometry. Based on time-of-flight methodology, reflections of light propagating along the fiber are analyzed for components of higher or lower wavelengths than the source light that provide information on changes to the optical fiber such as strain or temperature. Distributed sensing over many kilometers due to low power loss. Resistant to electromagnetic interference. Small diameter and lightweight. Broad bandwidth enables high-speed data transmission and high spatial resolution, and the signal can contain significantly more information than is possible with metallic conductors. Fragile optical fibers limit maximum strain to <1%. Sensitive to strain and temperature.	Sensing range up to 10 s of km Spatial resolution down to ~1 m. (trade-off between sensing range and spatial resolution). Maximum strain capability < 1%	[7,8,9]
FBG	Periodic modifications in the refractive index of the optical fiber are manufactured. This creates narrow-band reflections and discrete resonant frequencies. Changes in environmental conditions such as strain or temperature cause the distance between the periodic refractive index modifications to change, shifting the resonant frequencies. This shift is analyzed to infer the change in environmental conditions that has occurred. Many of the same attributes as DOF technology are still based on optical fibers. Can be embedded into composite fiber structures. Capable of single-point or multi-point sensing. Approximately 13–14 sensors can be multiplexed on a single fiber. Quasi-distributed as multiplexed Bragg gratings are spaced out along a fiber optic length. Fragile optical fibers limit maximum strain to <1%. Sensitive to strain and temperature.	Sensing range up to 10 s of km. Spatial resolution of ~2 mm. Maximum strain capability < 1%.	[2,10,11,12]
CCTDR	Time domain reflectometry. A time-of-flight measurement based on reflected signals from a coaxial cable. Defects along the line create new reflections. Coaxial cables are robust structures that can withstand rough environments. Only relatively large-scale deformation/defects will generate a new reflection that can be deflected. Lower strain events cannot be monitored. Coaxial cable heavier compared to fiber optics.	Sensing range ~10 s–100 s m	[13]
CCFPI	A pair of partial reflectors created by a change in cable impedance are manufactured on the cable. Reflections interfere to create a series of resonant frequencies that can be tracked. If the distance between the pair of partial reflectors changes due to strain or thermal expansion, the frequency of the resonances shift. Option for single-zone or fully distributed sensing. Coaxial cable is a robust structure. Further work to be carried out on the longevity and reliability before employed in the field. Coaxial cable heavier compared to fiber optics.	Maximum reported sensing range ~1m. Spatial resolution reported ~order of cm’s. Large strain capability (>5%).	[14,15,16]
Electrical resistance strain gauges	The resistance of a wire is dependent on the length and cross-sectional area. When a strain is applied to a wire, these dimensions change, altering the resistance which can be measured. Very established technology, low cost, and can be wired for temperature compensation in a bridge configuration. Point sensors. Installing many strain gauges results in complex wiring arrays. Impractical for measuring long distance structures.	3–5% maximum strain	[17]
RFID antenna	A rectangular micro-strip patch antenna constructed of a metallic patch on a substrate with a ground plane has a resonant frequency. If strain is applied to patch, changing the length of the structure, the resonant frequency shifts. A handheld stand-alone RFID reader is used to measure this change from which the change in strain can be inferred. Wireless, low-cost. Each RFID tag has a unique identifier so strain measurements can easily be linked to an exact location. Must be accessible for a reader to interrogate the device. Point sensors. Does not provide continuous monitoring.	Maximum strains of up to 10 s of %.	[18,19,20]

**Table 2 sensors-25-00650-t002:** Performance of distributed and quasi-distributed fiber optic sensing techniques (adapted from [10]).

Sensing Technology	Transducer Type	Sensing Range	Spatial Resolution	Main Measurands	Single-Ended Monitoring
Raman OTDR	Distributed	1 km 37 km	1 cm 17 m	Temperature	NO
Brillouin OTDR	Distributed	20–50 km	~1 m	Temperature and Strain	YES
Rayleigh OFDR	Distributed	50–70 m	~1 mm	Temperature and Strain	YES (needs a reference fiber)
Fiber Bragg Grating	Quasi-distributed	~100 channels	2 mm (Bragg length)	Temperature, Strain, and Displacement	YES

**Table 3 sensors-25-00650-t003:** Summary of work on modified coaxial cables to enhance TDR capabilities.

Type of Modification	Purpose of Modification	Citation
Dielectric material selection	Increased sensitivity to strain	[48]
Topology of outer conductor	Increased sensitivity to strain	[7,9,47,71,72]
Intentional creation of impedance changes	Tracking point to measure distance changes/act as ‘location tags’	[61]
Inclusion of small in-line sensors	Increase sensitivity to strain, or increase functionality of cable	[53,73]

**Table 4 sensors-25-00650-t004:** Summary of published information on development of coaxial cable strain/displacement sensors.

Coaxial Cable Sensor Type	Application	Cable Type	Partial Reflector	Sensor Active Length	Frequency	Key Results	Reference
CCFPI	Axial strain sensor	RG58	Holes	70 mm	~3.5 GHz	−3.3 kHz/µε	[46]
CCFPI	Axial strain sensor	Not stated	Not stated	Not stated	~1.9 GHz	22.5 kHz/µε	[14]
CCFPI	Torsion sensor (cascaded FPIs)	RG58	Crimping metal ferrules	227 mm (wrapped)	~4.2 GHz	1.834 MHz (rad/m)^−1^	[84]
CCFPI	Embedded in GFRP for core in steel stranded cable	SF047	Crimping metal ferrules	200 mm	~3 GHz	−3.7 kHz/µε	[45]
CCFPR	Measure lateral displacements	Bespoke structure	Metal post and metal plate	75 mm	~1 GHz	Lateral position resolutions measured to order of 1nm	[98]
CCFPR	Strain, e.g., shrinkage strain	Bespoke structure	Metal post and gap/flange	80 mm	~0.6 GHz	Sensitivity of 2.5 GHz/mm—nanoscale precision	[101]
CCFPR	Displacement sensor	Bespoke structure	Metal post and metal cone	20 cm	~1.2 GHz	Displacement to resolution of 10 µm	[102]
CCFPR	Strain sensor for high-temperature environments	Bespoke structure	Metal inserts	11.8 cm	~1 GHz	Monitored thermal strain between 100 and 900 °C	[103]
CCBG	Axial strain sensor	RG58	Holes	1.408 m (22 periods)	~4.25 GHz	−3 kHz/µε	[21]
CCBG	Axial strain sensor	RG58	Holes	1 m (40 periods)	~4 GHz	−2.1 kHz/µε	[15]
CCBG	Strain sensor	Bespoke	Undulating dielectric	200 mm (10 periods)	~4.5 GHz	~3.075 kHz/µε	[118]
Irregular CCBG	Strain sensor	MIL-C_17	Holes	2 m	0–6 GHz	−5.068 kHz/µε	[88]

**Table 5 sensors-25-00650-t005:** Summary and comparison of strain–temperature deconvolution methods.

Temperature–Strain Decoupling Method	Sensor Technology on Which Method Has Been Tested	Description/Comments		Strain Sensitivity	Temperature Sensitivity	Reference
None	FBG	Baseline for understanding potential impact of temperature on strain measurements.		7.72 ppm/µε	−101.62 ppm/°C	[120]
None	CCFPI			For a CCFPI with thermal expansion dominated by the dielectric (PTFE of 150 ppm), a 1 °C shift would result in a strain measurement of 150 µε.		
Design preferential response to strain	CCFPI	In a CCFPI, select a dielectric material that demonstrates a greater change in response to strain and had reduced reaction to temperature. Topology of the coaxial cable designed to enhance response to strain. May only serve to reduce sensitivity to temperature, not eliminate it.		A coaxial cable with a silicon rubber dielectric (instead of PTFE) and a spiral wrapped outer conductor showed 15–80 times increase in sensitivity to strain. Silicon rubber has a modulus of elasticity approximately 10 times lower than that of PTFE, so on that basis, it would theoretically provide a 10-fold increase in sensitivity to strain.		[7]
Inclusion of uncoupled/relaxed sections along CCFPI	CCFPI	Along the length of a CCFPI, several sections could be left uncoupled to the structure under strain. These ‘relaxed’ sections should then only be subjected to temperature fluctuations, and analysis of these sections can be used to calibrate the strained sections to remove unwanted temperature effects. Relaxed sections used for temperature compensation are not co-located with the strain sensing sections and could therefore be subjected to different temperature fluctuations, so calibration to eliminate temperature effects may not be accurate.		In theory, an ‘inactive’ co-located section of a CCFPI should provide full temperature compensation, subject to precise material composition and manufacturing variations, resulting in slightly different temperature response between active and ‘inactive’ sections.		[14]
Wheatstone bridge configuration	Electrical resistance strain gauges	Arrange multiple sensors in a half or full Wheatstone bridge configuration. One branch of the bridge monitors strain; other branches aid temperature compensation. A full Wheatstone bridge offers a more complete solution, but this does involve a greater number of sensor devices.		For a single electrical resistance strain gauge, sensitivity to strain is approximately 2 ppm/µε, and sensitivity to temperature is approximately 150 ppm/°C. A full Wheatstone bridge should eliminate the temperature response.		[119,127]
Active/dummy section	FBG	One active strain sensor, coupled to structure to be monitored, works with a ‘dummy’ strain sensor that is not coupled for strain monitoring, and therefore solely tracks temperature fluctuations. Active and dummy sections could be co-located for optimum temperature compensation.		In theory, a ‘dummy’ co-located section of an FBG should provide near-full-temperature compensation (subject to precise material composition and manufacturing variations resulting in slightly different temperature response between active and dummy sections).		[40]
FPI based on a bubble	Fiber optic	Bubble filled with air forms an FPI cavity. The low thermal conductivity of air reduces impact of temperature on strain measurements.	Including as a baseline for comparison with Vernier method.	7.75 ppm/µε 6.0 ppm/µε	1.2 ppm/°C 1.1 ppm/°C	[120]
Two parallel FPIs and Vernier effect	Fiber optic	Using two FPI fiber optic sensors in parallel. One active and sensing strain and one acting as a reference. The joint parallel response from the two sensors is used for analysis, with the Vernier effect aiding to enhance the sensitivity to strain.		−14.9 ppm/µε	0	[120]

## Data Availability

No data were generated in this work.

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
