# Peer review of "Coaxial Cable Distributed Strain Sensing: Methods, Applications and Challenges"

_sensors, 2025, doi:10.3390/s25030650_

Round 1
Reviewer 1 Report
Comments and Suggestions for Authors
The innovation of this paper lies in its relatively systematic study of the theoretical foundation, key technologies, and practical applications of coaxial cable-based distributed strain sensing. At the same time, by optimizing manufacturing processes, material selection, and signal processing methods, the feasibility and accuracy of this technology have been confirmed, providing new solutions for engineering fields such as structural health monitoring.
- The paper mentions that strain measurements are susceptible to temperature interference and suggests using data-driven analytical methods. However, it does not provide specific mathematical models or experimental validation to demonstrate how data-driven methods can effectively separate strain and temperature effects. It is recommended to include experimental studies focused on the temperature-strain coupling issue, propose a specific decoupling model, and compare the effects of different decoupling techniques.
- Although the paper mentions some practical applications (such as bridges, mines, and wind power equipment), experimental verification is mainly focused on laboratory environments, and the long-term performance in real engineering environments has not been fully assessed. It is suggested to extend the scope of experimental validation, such as conducting comprehensive tests on performance under different environmental conditions (high humidity, high pressure, high temperature). Additionally, the study should include real-world application cases, such as the actual deployment and long-term data analysis for bridge or tunnel monitoring.
- The paper mentions the advantages of signal processing technologies such as Joint Time-Frequency Analysis (JTFA), but does not provide detailed descriptions of specific implementation methods or algorithm optimization. It is recommended to supplement detailed signal processing steps, such as how to design and implement key parameters for sliding time windows, Fourier transforms, or other frequency-domain analysis methods.
- Although the paper compares the advantages and disadvantages of optical fibers and coaxial cables, it does not discuss the performance of other potential alternative technologies (such as wireless sensor networks or MEMS sensors). It is suggested to add comparative analyses with other sensing technologies to present the advantages and limitations of coaxial cable-based distributed sensing from a broader perspective.
Author Response
Dear Reviewer, please find attached authors feedback as PDF.

Reviewer 2 Report
Comments and Suggestions for Authors
The authors conducted a complete review of the current state of the art of distributed sensing of strain gauges based on coaxial cables. Distributed, quasi-distributed and point methods of strain gauge sensing based on fiber-optic sensors and based on coaxial cables were considered; the main advantages and disadvantages of each method were highlighted.
However, I would like to emphasize a few remarks on the paper:
Remarks on the text
1. I consider that, for better understanding of the material by the audience, it is worth adding to the review a description of the basic principles of each of the methods considered.
2. In the abstract on line 13-14 the disadvantage of distributed fiber-optic sensors “...their use is limited to lower strain applications due to the fragile nature of silica fiber” is given, but the paper does not compare the maximum range of elongation detection with coaxial cables. Also the paper does not mention the main advantage of fiber-optic sensors - wide bandwidth, which plays an important role in monitoring tasks.
3. In the section “1. Introduction” on line 47 the abbreviation ‘risk-based inspections (RBI)’ is given, but it is not used anywhere else in the text. I suggest removing the abbreviation.
4. In section “2. Background on fiber optic distributed sensing” on line 106-107 the expression “As a very basic description fiber optical sensing works by sending a pulse of light along a fiber and collecting the reflected light signal” is given, which is not quite correct - pulse sensing methods refer only to TDR and FDR methods, not to all fiber optic sensors.
5. In section “2.1. Time domain reflectometry in fiber optic sensing” on line 140-141 the abbreviation “Optical Time Domain Reflectometry (OTDR)” is given, but this abbreviation was already given on line 127. I suggest removing the abbreviation.
6. In “2.2.2 Grating based sensors” on line 203 and following, the abbreviation “FBG” is used, but the abbreviation is not provided in the text. I suggest to add a deciphering of the abbreviation.
7. In section “2.2.3. Interferometric Sensors” on line 222, the abbreviation “Surveillance d'Ouvrages par Fibres Optiques (SOFO)” is given, but is not used anywhere else in the text. I suggest removing the abbreviation.
8. In “2.2.3 Interferometric Sensors” on line 227, the abbreviation “distributed optical fiber sensors (DOFS)” is given, but this abbreviation was already given on line 118. I suggest removing the abbreviation.
9. In “3.1 Coaxial Cable Time Domain Reflectometry” on line 286, the abbreviation “time domain reflectometry (TDR)” is given on line 286, but the abbreviation was already given on line 112. I suggest removing the abbreviation.
10. In “3.1 Coaxial Cable Time Domain Reflectometry” on line 329, the abbreviation “Coaxial cable time domain reflectometry (CCTDR)” is given, but the abbreviation was already given on line 284. I suggest removing the abbreviation.
11. In “3.1.1 TDR using unmodified coaxial cables” on line 350, the abbreviation “time domain reflectometry (TDR)” is given, but the abbreviation was already given on line 112. I suggest removing the abbreviation.
12. In “3.1.2 TDR using modified coaxial cables” on line 539, the abbreviation “Spread Spectrum Time Domain Reflectometry (SSTDR)” is given, but the abbreviation is not used anywhere else. I suggest removing the abbreviation.
13. In “3.1.2 TDR using modified coaxial cables” on line 539, the abbreviation “Noise Domain Reflectometry (NDR)” is listed, but the abbreviation is not used elsewhere. I suggest removing the abbreviation.
14. In section “3.2 Coaxial Cable Frequency Domain Reflectometry” on line 577, the abbreviation “coaxial cable frequency domain reflectometry (CCFDR)” is listed, but the abbreviation is not used elsewhere. I suggest removing the abbreviation.
15. In “3.2.1 Interferometric coaxial cable frequency domain reflectometry” on line 597, the abbreviation “coaxial cable Fabry-Perot interferometry (CCFPI)” is given, but this abbreviation was already given on line 594. I suggest removing the abbreviation.
16. In “3.2.1 Interferometric coaxial cable frequency domain reflectometry” on line 654, the abbreviation “strain-tolerated Fabry-Perot interferometer (CCFPI)” is given on line 654, which is the same as the abbreviation “coaxial cable Fabry-Perot interferometry (CCFPI)” given on line 594. I suggest changing or removing the abbreviation.
17. In “3.2.1 Interferometric coaxial cable frequency domain reflectometry” on line 672, the abbreviation “CCBG” is given on line 672 and the abbreviation is explained later on line 894. I suggest adding the abbreviation on line 672 and removing the abbreviation on line 894.
18. In “3.2.1 Interferometric coaxial cable frequency domain reflectometry” on line 679, the abbreviation “linear variable differential transformer (LVDT)” is given, but this abbreviation is not used anywhere else. I suggest removing the abbreviation.
19. In “3.2.2 Grating-based coaxial cable frequency domain reflectometry” on line 931, the abbreviation “civil structure health monitoring (CSHM)” is given, but the abbreviation is given earlier on line 33. I suggest removing the abbreviation.
20. In “3.2.2 Grating-based coaxial cable frequency domain reflectometry” on line 977, the abbreviation “coaxial cable sensor (CCS)” is given, but is not used anywhere else. I suggest removing the abbreviation.
21. In “5. Conclusions” on line 1083, the abbreviation “civil structure health monitoring (CSHM)” is given on line 1083, but the abbreviation is given earlier on line 33. I suggest removing the abbreviation.
22. Under “5. Conclusions” on line 1102, the abbreviation “distributed optical fiber sensor (DOFS)” is given on line 1102, but the abbreviation is given earlier on line 118. I suggest removing the abbreviation.
23. All definitions that use two last names (e.g., Fabry and Perot) should be written with a middle hyphen (en dash), e.g., Fabry-Perot and Mach-Zehnder.
24. All expressions must be written in the same style, the styles of expressions (3)-(6) are different from expressions (1) and (2).
Remarks on the illustrations
25. The drawings 1, 4, 7-15, 17, 18, 20-24, 26-30, 33 and 34 should be improved in quality.
Misprints
26. Many units of measurement are given in the paper, but they are given together with their numerical value. It is worth separating (add a space symbol) the numerical value and the unit of measurement in all cases.
27. On line 98, “acoustic, temperature, strain, among”, an extra space is added after “temperature” - it should be “acoustic, temperature, strain, among”.
28. On line 123, “spatial resolution [18] .”, an extra space is added after the source reference.
29. On line 143 “[2][22]”, source citations should be in a single pair of square brackets separated by a comma - “[2,22]”.
30. On line 147 “ing. The sensing...” an extra period character is added.
31. On lines 289, 291, and 296, the lower index of the variables Z and ε should be “0”, not “o”.
32. On line 306, “length[49] Initially” has a missing space character after “length” and a missing dot sign before “Initially”.
33. On line 318 “[11,55],,” an extra comma character is added.
34. On line 321 “[7,40,59],.” an extra comma character is added.
35. On line 339, “were deduced in in an effort” has an extra “in” added.
36. On line 340, “to geological applications[56].”, a space character is missing before the source citation.
37. On line 373, “change along a 14 meter” should be “change along a 14-meter”.
38. On line 405, “loading conditions [41,51,52] .” an extra space character is added before the period sign.
39. On line 457, “spiral-wrap configurations . This” has an extra extra space character added before the period character.
40. On line 496, the word “Figure” should not be broken from the number “12”.
41. On line 602, “illustrated in Figure 15;” I believe should be “illustrated in Figure 15:”.
42. In all cases where negative numerical values are used, such as on lines 626 and 667, the minus sign “-” should be used instead of the short dash “-”.
43. On line 644, “different results.(a) Resonant” omits a space character before “(a)”.
44. On line 655 “been developed [37]; making”, an extra “;” is added.
45. On line 754 “than x distance apart, where [77];” I think it should be “than x distance apart, where [77]:”.
46. Wherever the Celsius temperature designation is used, such as on lines 817 and 870, the sign “℃” should be used or the degree sign “°” and the symbol “C” should be used.
47. Wherever the designation “Q-factor” is used, such as on lines 899, 901, 905 and 937, it shall be written with a “-” rather than a space character.
Author Response

(The authors gave the same response as above.)

Round 2
Reviewer 1 Report
Comments and Suggestions for Authors
Accept